# Reward salience but not spatial attention dominates the value representation in the orbitofrontal cortex

Wenyi Zhang[1,2,3], Yang Xie[1,2,3] & Tianming Yang ⬤[1] ✉

The orbitofrontal cortex (OFC) encodes value and plays a key role in value-based decision-making. However, the attentional modulation of the OFC's value encoding is poorly understood. We trained two monkeys to detect a luminance change at a cued location between a pair of visual stimuli, which were over-trained pictures associated with different amounts of juice reward and, thus, different reward salience. Both the monkeys' behavior and the dorsolateral prefrontal cortex neuronal activities indicated that the monkeys actively directed their spatial attention toward the cued stimulus during the task. However, the OFC's neuronal responses were dominated by the stimulus with higher reward salience and encoded its value. The value of the less salient stimulus was only weakly represented regardless of spatial attention. The results demonstrate that reward and spatial attention are distinctly represented in the prefrontal cortex and the OFC maintains a stable representation of reward salience minimally affected by attention.

Imagine that you are in a bookstore trying to find an ideal gift for your young child. While you direct your search in the children's book section, an eye-catching poster of your favorite writer in the bestseller section would divert your attention and lead you to check out the books over there. While task demands or behavior context often lead to internally generated attention, commonly referred to as top-down attention, salient stimuli, either based on physical or reward salience, may capture attention via a bottom-up mechanism[1–6]. Attention from top-down and bottom-up sources may be integrated or compete against each other in the brain and affect decision making.

Previous studies suggested that attention may affect value-based decision making by modifying choice options' subjective value[7,8]. Moreover, attention, both in the overt form with gaze shifts and in the covert form without eye movements, has been shown to modulate the representation of value in the orbitofrontal cortex (OFC)[9–11], which is a key brain area involved in value-based decision making and adaptive behavior[12–16]. In addition, the OFC neuronal activities encoded each choice option's value alternately during value-based decision making, which might be related to attentional shifts[17,18]. Based on these studies,

it has been proposed that the OFC activity reflects the value of the attended item while multiple items are presented.

However, in these studies, the researchers inferred the location of attention either from reward or visual salience[11] or from gaze location[9,10]. Without clear behavior or neurophysiology markers of attention, these studies only provided indirect evidence for the proposed theory. Especially when the behavior task and the stimulus or reward salience direct attention to distinct locations, an experiment that allows one to measure the attention location directly is necessary for investigating how attention from different sources modulates the OFC's value encoding.

To this end, we trained two macaque monkeys to perform a visual detection task with a Posner cueing paradigm[19]. A pair of visual stimuli were presented simultaneously, and a cue indicated which of the stimuli would change its luminance. The monkeys had to direct their attention toward the cued stimulus to detect the luminance change better. In addition, the stimuli were well-trained shapes that were associated with different amounts of reward. They acquired different levels of salience through over-training and could capture attention via

[1]Institute of Neuroscience, Key Laboratory of Primate Neurobiology, Center for Excellence in Brain Science and Intelligence Technology, Chinese Academy of Sciences, Shanghai 200031, China. [2]University of Chinese Academy of Sciences, Beijing 100049, China. [3]These authors contributed equally: Wenyi Zhang, Yang Xie. ✉e-mail: tyang@ion.ac.cn

a bottom-up mechanism. We were able to verify that the monkeys directed their attention according to the cue based on both the monkeys' behavior and the dorsolateral prefrontal cortex (DLPFC) neuronal activities. However, contrary to the proposed theory, we observed that the value encoding in the OFC was dominated by reward salience, regardless of where the cue and the attention were. The results indicate that the OFC maintains a stable salience-based representation of value that is only weakly modulated by spatial attention.

## Results

### Behavioral task and subject performance

We trained two monkeys to perform a visual detection task (Fig. 1a, d). In this task, the monkeys were instructed to covertly pay attention to one of the two stimuli to detect a luminance change at a random delay. The stimulus with the luminance change was cued by a frame that was presented on the side opposite to the change location 200 ms before the onset of the stimulus pair. The monkeys were required to report the change by making a saccade toward an eye-movement target located above the fixation spot to get a juice reward. For monkey D, the cue was valid in 90% of the trials. In the remaining trials (invalid-cue trials), the change location was on the same side of the frame (un-cued location). Monkey G was also trained to detect the luminance change at the cued location (target trials: 80%). In addition, to encourage monkey G to direct its attention appropriately, we required it to ignore the luminance change at the un-cued location and not to make a response (distractor trials: 20%). In half of the distractor trials, another

luminance change would happen at the cued location after the distractor (distractor+target trials: 10%), and monkey G was rewarded for detecting the change. In the other half of the distractor trials (distractor-only trials: 10%), there was no luminance change at the target stimulus, and the monkey had to hold its fixation till the end of the trial to receive a reward.

The stimulus set included 5 pictures for each animal. Each picture was associated with a juice reward of a different size (Fig. 1a, d, inset). The reward for correct responses was randomly chosen between the rewards associated with the two stimuli. Therefore, the reward outcomes were only 50% certain. The initial frame cue, as well as the stimulus where the luminance occurred, did not convey information about which stimulus's reward would be delivered. We initially trained the monkeys with single-stimulus trials, in which only one stimulus was presented, and the frame always appeared on its opposite side. Its associated reward would be delivered for a correct response, and there was no ambiguity about the reward outcome.

For the ease of the discussion, between the two stimuli presented in each trial, we refer to the stimulus that is cued to have a luminance change as the cued stimulus, its value as cued value (CV). The other stimulus is referred to as the un-cued stimulus and its value as un-cued value (UCV). In addition, as the monkeys were over-trained with this task and were familiar with the stimuli, the stimuli acquired different levels of salience due to their reward associations: high salience for stimuli associated with larger rewards and low salience for stimuli associated with smaller rewards. Accordingly, we refer to the stimulus

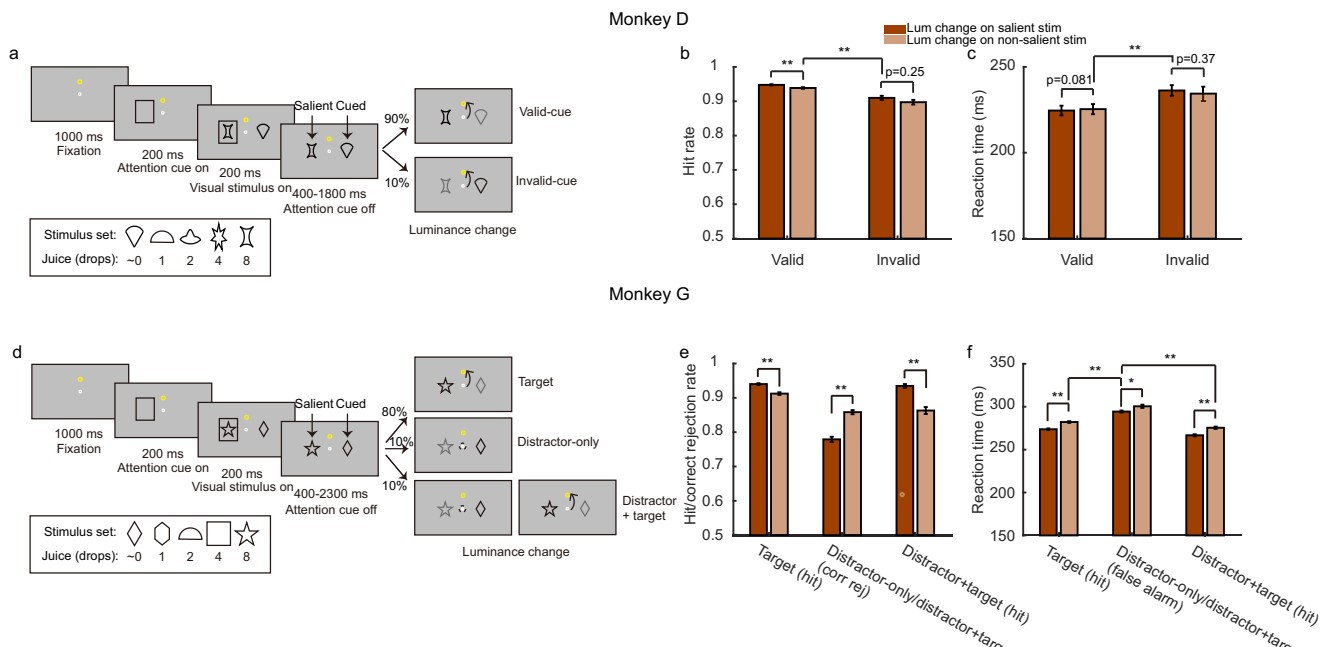

**Fig. 1 | Behavioral tasks and monkeys' performance. a** Monkey D had to detect when any of the two stimuli changed its luminance. The location of the luminance change was indicated by a cue (90% valid) that appeared on the side opposite to the luminance change. When the cue was invalid (10% of trials), the luminance change occurred on the same side of the cue. The monkey reported the luminance change by making an eye movement to the eye-movement target located above the fixation point. The stimuli were associated with different amounts of juice reward (inset). Random pairings of stimuli were selected for each trial. The stimulus with a larger associated reward is referred to as the salient stimulus, and the stimulus where attention should be direct to is the cued stimulus. The computer randomly selected one stimulus from the pair and delivered its associated reward for a correct response. **b** The proportion of correct responses when the cue was valid and invalid and when the luminance change occurred at the salient and the non-salient stimulus. **c** The reaction times when the cue was valid and invalid and when the

luminance change occurred at the salient and the non-salient stimulus. **d** Monkey G was required to detect the luminance change at the target location, indicated with the cue frame (target trials, 80%), and ignore the change at the un-cued location (distractor trials, 20%). In half of the distractor trials, another luminance change happened at the cued location after the distractor changed its luminance (distractor+target trials, 10%). The stimuli were associated with different amounts of juice reward (inset). **e** The proportion of correct responses in the three trial types and when the luminance change occurred at the salient and the non-salient stimulus. **f** The reaction time of detecting the target and the distractor luminance change and when the luminance change occurred at the salient and the non-salient stimulus. The error bars are SEM across sessions (monkey D: $n = 71$; monkey G: $n = 100$), * denotes $p < 0.01$, ** denotes $p < 0.001$, two-tailed Wilcoxon signed-rank test.

associated with a larger reward as the salient stimulus, and the other one as the non-salient stimulus. Their values are termed salient value (SV) and non-salient value (NSV). SV and NSV are the same in trials with a pair of identical stimuli.

Both monkeys learned the task and used the cue appropriately to direct their attention. When the cue was valid, monkey D were more accurate (valid: mean hit rate = 94.2%, SEM = 0.2%; invalid: mean hit rate = 90.3%, SEM = 0.4%. $p \ll 0.001$, two-tailed Wilcoxon signed-rank test. SEMs are across sessions) and responded faster (valid: mean reaction time (RT) = 224.5 ms, SEM = 2.9 ms; invalid: mean RT = 234.1 ms, SEM = 3.6 ms. $p \ll 0.001$, two-tailed Wilcoxon signed-rank test). Similarly, Monkey G accurately detected the luminance change at the target (target: mean hit rate = 92.2%, SEM = 0.3%; distractor+target: mean hit rate = 90.0%, SEM = 0.6%) and ignored the distractor's luminance change (distractor trials: mean correct rejection rate = 81.4%, SEM = 0.6%). The response latency of the target was shorter than that of the distractor in the false alarm trials (target responses in target trials: mean RT = 278.2 ms, SEM = 0.9 ms; target responses in distractor +target trials: mean RT = 271.2 ms, SEM = 1.1 ms; distractor responses: mean RT = 297.8 ms, SEM = 1.0 ms. $p \ll 0.001$ for the difference between the target response in either target or distractor + target trials and the distractor responses, two-tailed Wilcoxon signed-rank test).

The stimulus salience also affected the monkeys' performance, although to a much lesser degree than the cue. The monkeys were more likely to detect a luminance change of a salient stimulus than that of a non-salient stimulus (monkey D: valid salient: mean hit rate = 94.8%, SEM = 0.2%; valid non-salient: mean hit rate = 93.9%, SEM = 0.2%; monkey G: target salient: mean hit rate = 94.0%, SEM = 0.3%; target non-salient: mean hit rate = 91.2%, SEM = 0.4%. $p \ll 0.001$ for both monkeys, two-tailed Wilcoxon signed-rank test, Fig. 1b, e). Monkey G also responded faster when the target stimulus was the salient stimulus (target salient: 273.8 ms, SEM = 1.0 ms; target non-salient: 282.2 ms, SEM = 1.0 ms. $p \ll 0.001$, two-tailed Wilcoxon signed-rank test, Fig. 1f), while such an improvement was not seen in monkey D (Fig. 1c). These behavior improvements caused by reward salience were however much smaller than those caused by the cue (monkey D: valid non-salient: mean RT = 225.4 ms, SEM = 2.9 ms; invalid salient mean RT = 236.3 ms, SEM = 3.1 ms; monkey G: target non-salient: mean RT = 282.2 ms, SEM = 1.0 ms; distractor salient: mean RT = 294.4 ms, SEM = 1.2 ms. $p \ll 0.001$, two-tailed Wilcoxon signed-rank test, Fig. 1c, f). As the reward for correct responses was randomly chosen between the two stimuli, paying attention to the salient stimulus did not bring any behavior benefits to the monkeys or provide information about the expected reward. Correspondingly, the behavior of the monkeys indicated that their attention was dominated by the cue.

## DLPFC population encodes spatial attention shifts

In addition to the evidence from the monkeys' behavior, we recorded single-unit activity from the DLPFC neurons to further confirm that the monkeys' attention was directed to the cued location. A total of 406 DLPFC single units (monkey D: 240; monkey G: 166) from Walker's areas 8a, 8b, 46d, and 46 v (Supplementary Fig. 1) were recorded, and their activities were used to decode the location of attention[20,21].

One important feature of our task is that the frame serving as the cue was displayed on the side opposite to where the attention needed to be directed. The onset of the frame might first capture the monkeys' attention to a wrong location via a bottom-up process. Afterward, the monkeys had to shift their attention to the opposite side where the luminance change was expected. This attention shift was captured by many DLPFC neurons that encoded spatial attention location. In Fig. 2a we show such an example neuron. The neuron had larger responses initially when the frame appeared to the left of the fixation point (blue trace). After the cue offset, its response became greater in trials that had the cue frame on the right but the cued stimulus on the left (red trace). The neuron's responses reflected the attention shift from the

side of the frame to the opposite side where the luminance change was expected.

Population analyses confirmed that the DLPFC neurons encoded spatial attention location and reflected the attentional shift. To pool neurons with different spatial selectivity together, we considered the whole population of recorded neurons as a high-dimensional representation of task-related variables[17,22,23] and trained a decoder based on the linear discriminant analysis (LDA) algorithm with a pseudo neuronal ensemble to decode the frame location. The pseudo neuronal ensemble was composed in such a way that each neuron had the same number of left and right frame trials (see Methods). The decoders based on the LDA provided the posterior probabilities of the frame location.

We trained and tested the decoders with DLPFC neurons' responses using a sliding window (size: 25 ms, step: 10 ms) and plotted the decoders' performance as a heat map (Fig. 2b). When the decoders were trained and tested with the responses at the same time point, indicated by the diagonal in Fig. 2b, the cue location can be reliably decoded shortly after the onset of the attention cue (Fig. 2c, left) until after the luminance change and the animal's response (Fig. 2c, right). During the period from -200 ms after the stimulus onset until after the luminance change, the encoding of the attention location was stable so that the cross-temporal decoding performance was maintained (Fig. 2b). Furthermore, the cross-temporal decoder's performance revealed the attention shift after the shape onset. We trained a decoder with the neuronal responses 50–200 ms before the stimuli onset when only the frame but not the stimuli were on the screen. The decoder's performance was then tested with the responses at the other time points in a trial (Fig. 2d). The decoder's performance quickly dropped below the shuffled level after the stimulus onset and reached its minimum after the luminance change. Lower-than-shuffled performance indicates that the attention location shifted to the side opposite to where it was when the decoder was trained. This attention shift can be also observed in Fig. 2b in the dark blue regions—an indication of the performance below the chance level—where the decoders were trained and tested at different time points when the frame was presented and when the stimuli were presented.

The encoding of attention location in the DLPFC correlated with the monkeys' performance. We divided the trials by the monkeys' RT into two halves and tested the decoder's performance in each half of the trials. The decoder performed significantly better before the onset of the luminance in trials when the monkeys reacted faster to the luminance change, suggesting that the attention was more likely to be allocated correctly in those trials (Fig. 2e, Supplementary Fig. 2). In addition, we were able to decode attention in the correct trials better than in the error trials (Supplementary Fig. 3).

## OFC responses were dominated by the salient value

After verifying that the monkeys directed their attention to the proper location, we investigated how the OFC neurons' value encoding was modulated by attention.

We recorded the activities of 349 OFC neurons (monkey D: 212; monkey G: 137) from Walker's areas 11 and 13 (Supplementary Fig. 1). Many of them encoded stimulus values. An example neuron is shown in Fig. 3a, d, g, j. We first used trials with two identical stimuli to measure the neuron's value selectivity. In these trials, the monkeys should be certain of the reward they would get for a correct response, and our previous study indicated that the OFC neurons responded similarly to a pair of identical stimuli and to the same stimulus presented alone[11]. The example neuron encoded the value of the stimuli; its responses were larger when the value was larger (Fig. 3a). To study the neuron's responses to a pair of distinct stimuli, we first grouped the trials by SV. The neuron's responses reflected SV stably during the stimulus period well until after the luminance change (Fig. 3d). We calculated the neuron's firing rates between the stimulus onset and

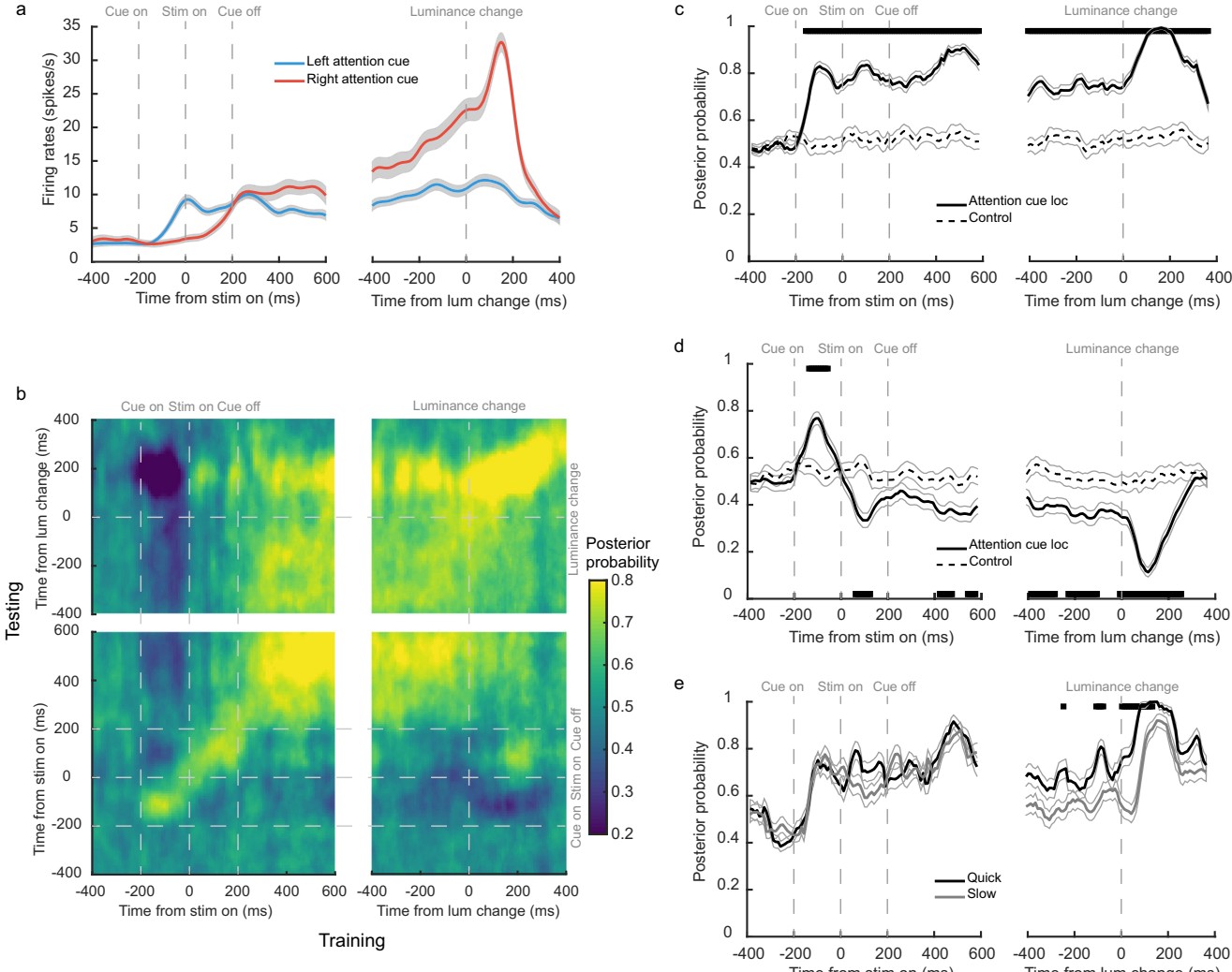

**Fig. 2 | DLPFC encoded spatial attention. a** The responses of an example DLPFC neuron. The red line indicates trials when the cue was on the right side and the blue line indicates trials when the cue was on the left side. The shaded areas around each line denote SEM across trials. **b** The posterior probability of the attention location decoded from DLPFC pseudo-population ensemble activities. Both the training and the testing of the LDA decoder used sliding windows of 25 ms at 10 ms steps. **c** The posterior probability of the attention location from the decoder that was trained and tested with the DLPFC neuronal responses at the same time (the diagonal line in **b**). Significance was assessed with two-tailed paired *t*-tests (*p* < 0.01 with FDR corrections for multiple comparisons), tested against the data with shuffled cue location labels. The black solid trace indicates the posterior probability calculated from the actual data. The black dashed trace indicates the posterior probability calculated from the shuffled data. The black segments at the top indicate when the actual data performed significantly better than the shuffled data. Thin gray lines represent SEM across trials. **d** Same as **b** except that the decoder was trained with the mean activities at 50–200 ms before the stimulus onset. The black segments at the bottom indicate when the performance was significantly lower than the control, indicating the flip of attention to the opposite side. Significance was assessed with two-tailed paired *t*-tests (*p* < 0.01 with FDR corrections for multiple comparisons). Thin gray lines represent SEM across trials. **e** The posterior probability of the attention location decoded from the fast (black line) and the slow trials (gray line). The trials were divided by the median reaction time in each neuron. Thin gray lines represent SEM across trials. Significance was assessed with two-tailed paired *t*-tests (fast versus slow, at *p* < 0.01 with FDR corrections for multiple comparisons).

the luminance change and plotted them against SV (Fig. 3g). The responses to SV were similar to the responses when a pair of the same stimuli were presented. The non-salient stimulus was largely ignored. In comparison, the neuron exhibited similar responses when the trials were grouped by CV and when the trials were grouped by the UCV (Fig. 3j). Both CV and UCV contributed to the neuron's responses similarly. The results suggested that the reward salience but not the cue dominated the example neuron's responses.

The OFC population exhibited the same trend. Among the 349 OFC neurons from which we recorded, 126 neurons were selective to value (Supplementary Table 1). We divided these neurons into positively tuned neurons and negatively tuned neurons based on their value tuning (see Methods). The positively tuned neurons had larger responses when the values were higher (Fig. 3b, 73 out of 126 value-

tuned neurons), and the negatively tuned ones had the opposite tuning (Fig. 3c, 53 out of 126 value-tuned neurons). Both groups of neurons encoded SV stably during the stimulus period (Fig. 3e, f). Their firing rates sorted by SV were similar to when a pair of the stimuli with the same value as SV were presented (Fig. 3h, i and Supplementary Fig. 4), and their responses did not distinguish between CV and NCV (Fig. 3k, l and Supplementary Fig. 5). The results were consistent across the two individual monkeys (Supplementary Fig. 6). Finally, to study the interaction between cue and reward salience, we grouped the trials by cued-SV, un-cued-SV, cued-NSV, and un-cued-NSV and plotted the OFC neurons' responses to each group condition. The responses were mostly divided between the salient and non-salient groups, while the cued and the un-cued groups exhibited similar responses, indicating little interaction between cue and reward salience. (Supplementary Fig. 7).

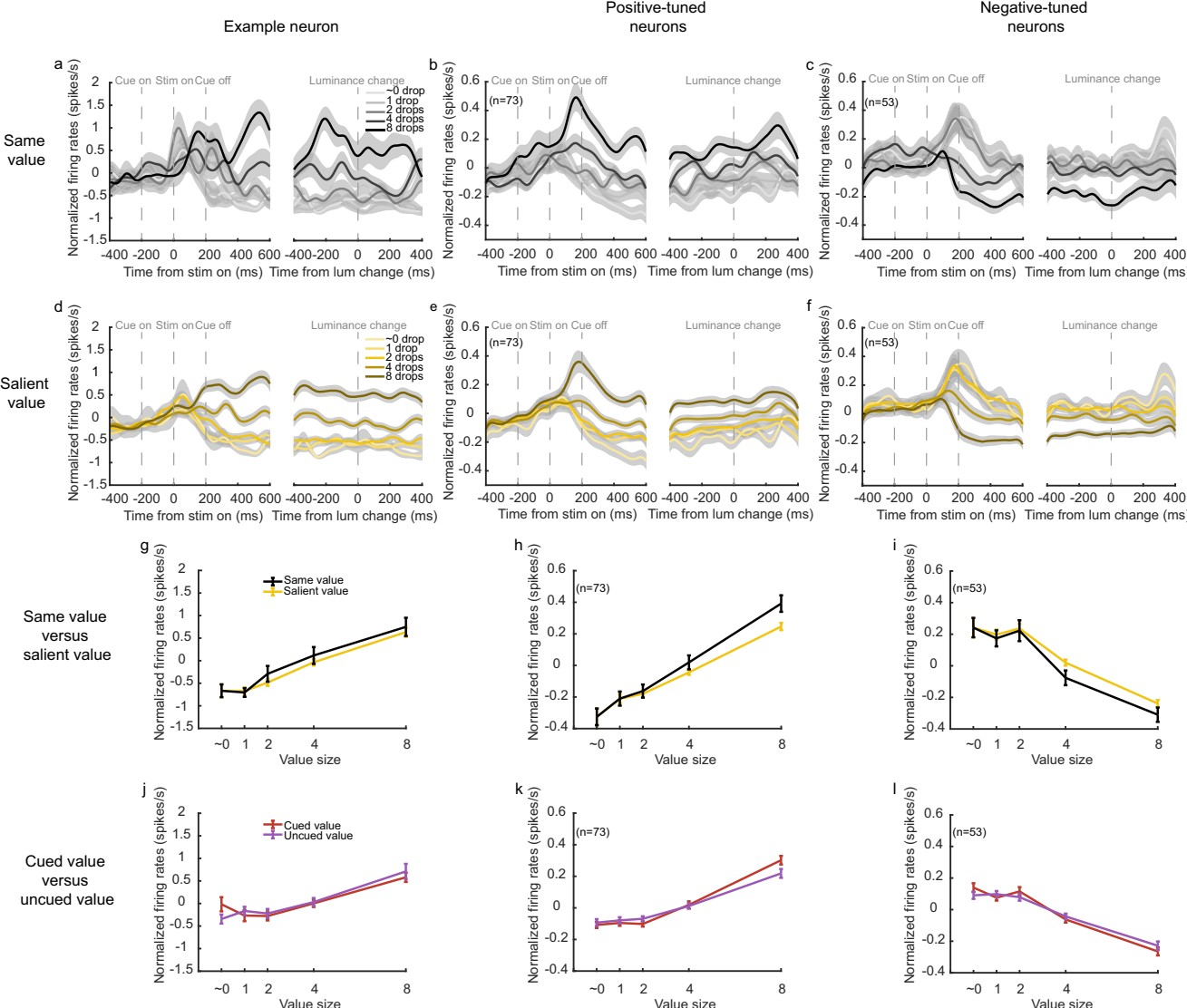

**Fig. 3 | Attentional modulation of the OFC responses. a** The responses of an example OFC neuron in trials when the same rewards were associated with the two stimuli, aligned to the stimulus onset (left) and the luminance change (right). The reward sizes are indicated by different shades of gray. Shading areas indicate SEM across trials. **b** The population response of the positively tuned OFC neurons ($n = 73$) in trials when the same rewards were associated with the two stimuli. Shading areas indicate SEM across neurons. **c** Same as **b**, but for the negatively tuned OFC neurons ($n = 53$). **d** The responses of the example neuron with the trials grouped by SV. The reward sizes are indicated by different shades of yellow. Shading areas indicate SEM across trials. **e** The responses of the positively tuned OFC neurons with the trials grouped by SV. The reward sizes are indicated by different shades of yellow. Shading areas indicate SEM across neurons. **f** Same as (**d**), but for the negatively tuned OFC neurons. **g** The average responses of the example neuron during the period between the stimulus onset and the luminance change. The black trace is based on the trials with the same-reward stimulus pairs.

The yellow trace is based on the trials grouped by SV. The error bars indicate SEM across trials. Two-way ANOVA (group: $F_{1413} = 0.79$, $p = 0.37$; value: $F_{4413} = 40.40$, $p \ll 0.001$). **h** The average responses of the positively tuned OFC neurons during the period between the stimulus onset and the luminance change. The black trace is based on the trials with the same-reward stimulus pairs. The yellow trace is based on the trials grouped by SV. The error bars indicate SEM across neurons. Two-way ANOVA (group: $F_{1724} = 3.22$, $p = 0.07$; value: $F_{4724} = 77.71$, $p \ll 0.001$). **i** Same as (**h**), but for the negatively tuned OFC neurons. Two-way ANOVA (group: $F_{1524} = 1.97$, $p = 0.16$; value: $F_{4524} = 46.03$, $p \ll 0.001$). **j** Same as **g**, but trials are grouped by CV (red) and NCV (purple). Two-way ANOVA (group: $F_{1690} = 0$, $p = 0.99$; value: $F_{4690} = 21.04$, $p \ll 0.001$). **k** Same as **h**, but trials are grouped by CV (red) and NCV (purple). Two-way ANOVA (group: $F_{1724} = 0.16$, $p = 0.68$; value: $F_{4724} = 104.56$, $p \ll 0.001$). **l** Same as **i**, but trials are grouped by CV (red) and NCV (purple). Two-way ANOVA (group: $F_{1524} = 0.01$, $p = 0.92$; value: $F_{4524} = 88.54$, $p \ll 0.001$).

We further quantified the contributions of SV and NSV to the OFC neuronal responses with a linear regression model that contained both value variables as well as a binary term that indicated the frame location. We calculated the coefficient of partial determination (CPD) to measure how much variance was explained by each variable (see Methods). The average CPD of SV rose significantly above the baseline shortly after the stimulus onset (Fig. 4a). In contrast, the CPD of NSV remained low and was not significantly different from the baseline in most time bins. The paired *t*-tests that compared the CPDs between SV

and NSV indicated that the OFC neurons encoded the SV much better in almost all the time bins after the stimulus onset ($p < 0.005$ with FDR correction for multiple comparisons). The frame location was transiently encoded when the frame was presented and after the luminance change but was not significantly encoded during most of the stimulus presentation period.

This SV dominance is also observed at the level of individual neurons. We calculated each neuron's average responses during the stimulus period and used them to determine the CPD for SV and NSV

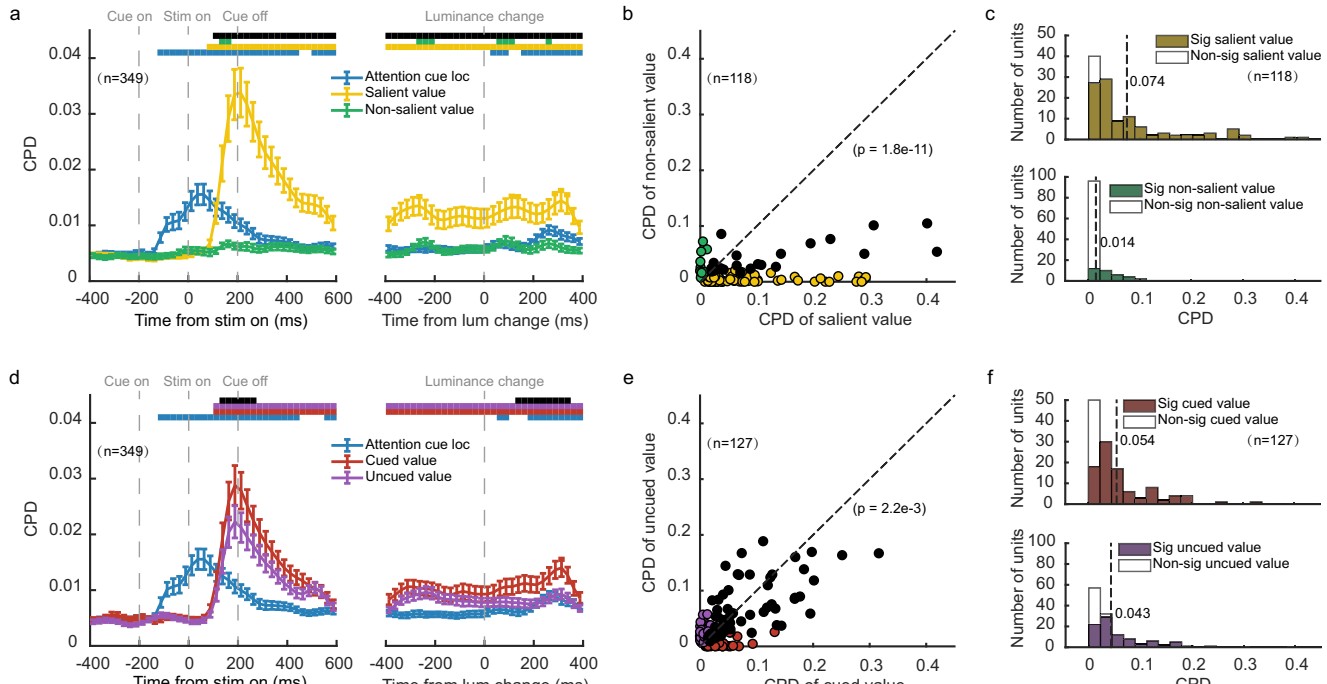

**Fig. 4 | OFC neuronal responses dominantly encoded the salient value. a** OFC neuron ($n = 349$) firing rates were regressed against attention cue location, SV, and NSV. Plotted is the time course of the population average coefficients of partial determination (CPD). Significance was assessed with two-tailed paired $t$-tests ($p < 0.005$, with FDR corrections for multiple comparisons) compared to a baseline computed with the CPD between 0 and 200 ms before the cue onset averaged across different regressors. The blue, yellow, and green bars at the top indicate the significant CPDs of cue location, SV, and NSV, respectively, and the black bar indicates significant differences between SV and NSV. The error bars indicate SEM across neurons. **b** The CPDs of SV against the CPDs of NSV for individual value-selective OFC neurons ($n = 118$). The yellow data points are the neurons with non-zero coefficients for SV only, the green data points are the neurons with non-zero coefficients for NSV only, and the black data points are the neurons that encoded both SV and NSV (one-sample $t$-test, $p < 0.05$, without multiple comparisons). Two-tailed paired $t$-test was conducted *to* compare the mean CPDs of SV and NSV of all the OFC neurons ($n = 349$). **c** Top: the distribution of the CPDs for SV of the value-selective OFC neurons ($n = 118$). Bottom: the distribution of the OFC neurons' CPDs for NSV. Vertical dashed lines indicate the mean. Filled bars indicate significant neurons. **d** Same as **a**, but for CV and UCV ($n = 349$). **e** Same as **b**, but for CV and UCV ($n = 127$). **f** Top: the distribution of the CPDs for CV of the value-selective OFC neurons ($n = 127$). Bottom: the distribution of the OFC neurons' CPDs for UCV. Vertical dashed lines indicate the means. Filled bars indicate significant neurons.

for each value-selective neuron (Fig. 4b). The data points of the neurons spread along the axis of the CPD for SV (SV: 105/349 significant, $p < 0.05$, linear regression). Their CPDs for NSV, on the other hand, clustered within a small range (NSV: 34/349 significant, $p < 0.05$, linear regression). The distribution of the neurons' CPD for SV was much more extended in the range than that for NSV, and the mean of the CPDs for the SV was larger than that of UCV ($t = 6.95$, $p = 1.8e-11$, two-tailed paired $t$-test) (Fig. 4c).

By contrast, when we regressed the OFC neuronal responses against CV and UCV, we obtained overall similar CPDs, suggesting that CV and UCV contributed similarly to the responses (Fig. 4d). Other than a short period after the stimulus onset and another one after the luminance change, the difference between CV's and UCV's CPDs was not significant during most of the stimulus period. During the period when the CPD of CV was larger, there were extra visual inputs, the frame initially and the luminance change in the end. When we plotted the neurons' CPDs for CV and those for UCV, they tended to lie closely to the diagonal (Fig. 4e). Nevertheless, the CPDs for CV were slightly but significantly higher than those for NCV ($t = 3.09$, $p = 2.2e-3$, two-tailed paired $t$-test). These results indicate that the cue and the spatial attention weakly and transiently modulated the neurons' value tuning. These results were again consistent across the two individual monkeys (Supplementary Fig. 8). Further regression analyses based on the four combinations of SV, NSV, CV, and NSV also point to the same conclusion (Supplementary Fig. 9).

Finally, to confirm that the weak attentional modulation was not due to a washout effect by the many trials in which CV and UCV had

similar values, we studied the trial conditions in which we would expect the largest modulation on the neurons' responses by spatial attention. According to our previous study[11], the cue-induced modulation might be most evident when the cue directed the attention toward the non-salient picture when the SV was the greatest (8 drops of juice). In this condition, the cue shifted the attention away from the most salient picture. This would provide us an opportunity to observe any potential modulation of the neurons' responses by the cue. Therefore, in these trials, we compared the neurons' responses when the cue directed the attention toward the 8-drops-of-juice cue (CV = 8) and when the cue directed the attention away from it (CV = -0, 1, 2, 4, or 8 and UCV = 8). Again, we looked at the positively and negatively tuned OFC neurons separately. When the attention was on the 8-drops-of-juice stimulus, the cue and the reward salience were consistent, and the OFC neurons largely ignored the value of the less salient stimulus (linear regression, H₀: slope = 0: $p = 0.44$, positively tuned neurons; $p = 0.30$, negatively tuned neurons). The responses were highest for the positively tuned neurons and lowest for the negatively tuned ones, both similar to their responses when a pair of 8-drops-of-juice stimuli were presented (Fig. 5a, b). When the cue directed the attention away from the 8-drops-of-juice stimulus, it lowered the responses of positively tuned neurons very slightly ($F_{1723} = 3.74$, $p = 0.05$) and failed to increase the responses of the negatively tuned neurons ($F_{1524} = 1.18$, $p = 0.28$). A complete shift of responses toward the less salient stimulus would have produced responses similar to those when a pair of the less salient stimuli were presented (black trace in Fig. 5a, b). Attending away from the most salient stimulus did not abolish the

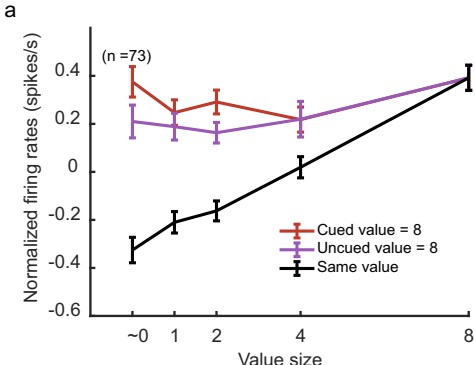
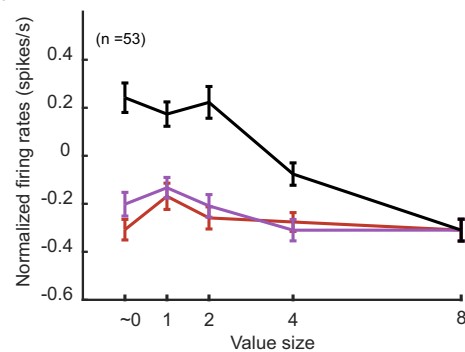

**Fig. 5 | Spatial attention failed to switch the value encoding in the OFC to the non-salient stimulus. a** The positively tuned OFC neurons' ($n = 73$) responses to an 8-drops-of-juice stimulus (SV) paired with a stimulus associated with ~0, 1, 2, 4, or 8 drops of juice (NSV). The red line denotes trials when the attention was toward the 8-drops-of-juice stimulus, and the purple line denotes trials when the attention was away from the 8-drops-of-juice stimulus. A two-way ANOVA (attention location: $F_{1723} = 3.74$, $p = 0.05$; value: $F_{4723} = 3.45$, $p = 8.4e{-}3$) indicates a subtle but significant attentional modulation. A complete switch of the value encoding to NSV would produce responses close to the black line, which are the neurons' responses to the trials of a pair of stimuli with the same reward. All error bars indicate SEM across neurons. **b** Same as **a**, but for the negatively tuned OFC neurons ($n = 53$). Two-way ANOVA with group (CV = 8/UCV = 8) and value was performed (attention location: $F_{1524} = 1.18$, $p = 0.28$; value: $F_{4524} = 3.68$, $p = 5.8e{-}3$).

dominant representation of SV by the OFC, and the overall neuronal responses were very similar regardless of whether the cue was on the most salient stimulus or not. The results were confirmed in the two individual monkeys (Supplementary Fig. 10).

### DLPFC's value encoding was also dominated by reward salience

It has been suggested that the DLPFC sits downstream of the OFC in the processing of value information. We wonder whether the dominance of the value encoding in OFC by reward salience may be inherited by the DLPFC. On the other hand, the DLPFC has a strong spatial attention signal that is lacking in the OFC (Supplementary Fig. 11). One may expect that spatial cue and attention may play a more substantial role in DLPFC's value encoding.

We carried out the analyses on DLPFC neuronal responses parallel to the analyses on the OFC. The results, which are summarized in Fig. 6 and Supplementary Figs. 12–14, indicated that the DLPFC's value encoding was similarly dominated by reward salience, and spatial attention did not affect DLPFC's value encoding significantly. The regression analyses similar to Fig. 4 but done to the DLPFC neuronal responses reveal that the population average CPD of SV was significantly above the baseline right after the stimulus onset and lasted until the end of the trials (Fig. 6a). The CPD of NSV was low, although initially significant, and did not last until the luminance change. Importantly, the CPDs of SV were consistently larger than the CPDs of NSV during the whole stimulus period (Fig. 6a). The CPDs of individual units for SV were also significantly greater than those for NSV (Fig. 6b, c, $t = 7.27$, $p = 1.9e{-}12$, two-tailed paired $t$-test). The dominance of SV was strong in monkey G (Supplementary Fig. 13g–i). Monkey D's DLPFC neurons encoded SV relatively weakly, yet the encoding was still stronger than that of NSV (Supplementary Fig. 13a–c). In contrast, the encodings of CV and UCV in the DLPFC were similar, both when in the combined data (Fig. 6d–f, $t = 1.56$, $p = 0.12$, two-tailed paired $t$-test) and in the individual monkeys (Supplementary Fig. 13d–f, j–l). Therefore, despite that a strong spatial attention signal is present in the DLPFC (blue traces in Fig. 6a, d), its modulation of the value encoding was weak.

### Comparison between OFC and DLPFC

Finally, to compare the representations of spatial attention and value in the OFC and the DLPFC, we plotted the CPD for attention cue location against the CPD for SV for each neuron from the two areas (Fig. 7). The CPDs were calculated from regression models similar to those used in Figs. 4b, 6b, but the average neuronal responses were computed with the period between the frame cue offset and the luminance change, during which spatial attention was mostly driven by the top-down mechanism and on the opposite side of the cue. The CPDs of OFC neurons are largely distributed along the SV axis, while the CPDs of DLPFC neurons were more evenly distributed along both axes. Furthermore, for the OFC neurons, the distribution of the neurons' difference between the two CPDs strongly skewed toward SV (Fig. 7 insert, mean $\Delta$CPD = −0.072, $p = 1.7e{-}9$, two-tailed paired $t$-test). In contrast, the DLPFC neurons did not exhibit such a difference (mean $\Delta$CPD = −0.0031, $p = 0.43$, two-tailed paired $t$-test). The analyses confirmed that SV dominated the OFC responses, while both SV and spatial attention were well represented in the DLPFC. Similar results were observed in individual monkeys (Supplementary Fig. 15).

## Discussion

Here, we demonstrated that while both the behavior and the DLPFC neuronal responses indicated that the monkeys directed their attention to the cued location, the OFC neuronal responses were nevertheless dominated by stimulus reward salience and only weakly affected by spatial attention. The results argue against the previously proposed theory that attention serves as a selection mechanism for the OFC's value encoding when multiple items with different value associations are presented[11].

The dominance of reward salience that we observed cannot be attributed to spatial attention, decision making, or eye movements. First, unlike reward salience, which was constant in a trial, the cue that indicated the luminance change was on the opposite side of the luminance change, and the spatial attention needed to switch sides. Therefore, spatial attention could be clearly identified with the switch of location and be distinguished. Second, the monkeys had to report their decision by making a saccade to a target above the fixation point. The motor preparation signals were thereby dissociated from the attention and salience signals. Third, the reward that the monkey received was randomly chosen between the two stimuli. Neither the CV nor the SV was the expected reward outcome. Lastly, no decisions were guided by the values of the stimuli, so potential attention shifts accompanying the value-based decision-making process were minimized with the current behavior paradigm.

Spatial attention in our paradigm was dictated by the attention cue via a top-down mechanism, while reward salience may create attention from a bottom-up subcortical circuitry, which includes

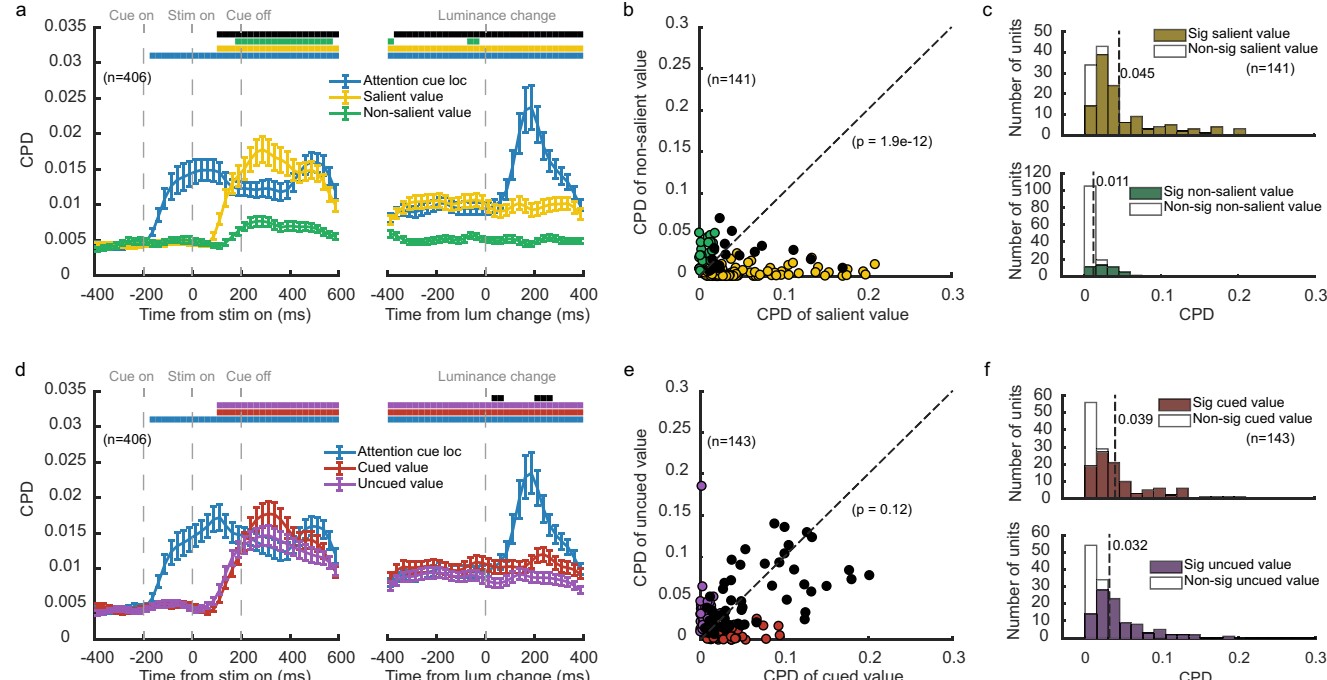

**Fig. 6 | DLPFC neuronal responses were dominated by the salient value.**
**a** DLPFC neuron ($n = 406$) firing rates were regressed against attention cue location, SV, and NSV. Plotted is the time course of the population average coefficients of partial determination (CPD). Conventions as in Fig. 4a. **b** The CPDs of SV against the CPDs of NSV for individual value-selective DLPFC neurons ($n = 141$). Conventions as in Fig. 4b. **c** Top: the distribution of the CPDs for SV of the value-selective DLPFC neurons ($n = 141$). Bottom: the distribution of the DLPFC neurons' CPDs for NSV. Vertical dashed lines indicate the mean. Filled bars indicate significant neurons. **d** Same as (**a**), but for CV and UCV ($n = 406$). **e** Same as **b**, but for CV and UCV ($n = 143$). **f** Top: the distribution of the CPDs for CV of the value-selective DLPFC neurons ($n = 143$). Bottom: the distribution of the DLPFC neurons' CPDs for UCV. Vertical dashed lines indicate the means. Filled bars indicate significant neurons.

superior colliculus, pulvinar, ventral midbrain, and amygdala[24,25]. Although reward salience is distinct from the kind of bottom-up attention based on physical salience, interestingly, we also observed bottom-up physical salience signals in the OFC. First, the OFC neurons robustly, although only briefly, encoded the onset of the frame (Fig. 4a, d, blue trace), which was a salient bottom-up visual signal. Second, there was a slightly stronger representation of the cued stimulus than that of the un-cued after the luminance change (Fig. 4d), which, although subtle, might also capture the attention via a bottom-up mechanism. Therefore, while the OFC activities were minimally affected by spatial attention from the top-down source, influences from the bottom-up sources, no matter whether they were based on physical or reward salience, were readily reflected in the OFC.

The dominance of reward salience, which was almost all-or-none, is in stark contrast to how neurons in the visual cortex are modulated by spatial attention[26]. The responses of the visual neurons to multiple stimuli presented in their receptive field can be well described with normalization models that perform a weighted average of the responses to each stimulus presented alone. Attention modulates how the responses are weighted by adding bias toward the attended stimulus, and the neurons' stimulus preference plays an important role. Whether attended or not, a preferred stimulus presented in a neuron's receptive field has a significant influence on its responses[27]. OFC neurons, however, only encode the value of the most salient stimulus. That is true even for the negatively tuned neurons, which prefer the non-salient stimulus. Less salient stimuli are largely ignored by OFC neurons, and spatial attention, when directed toward a less salient stimulus, does not enhance its encoding very much in the OFC.

One might argue that the weak representation of NSV in the OFC was because of the task design, in which the reward for a correct response was randomly chosen between the two stimuli. Yet, the value that was relevant to the monkeys in this task—the mean value of the two stimuli—was also not represented by the OFC. For that to be true, one would expect that SV and NSV be similarly represented. Yet, the findings of the dominance of SV encoding and the weak attentional modulation may not generalize to other species. Notably, the OFC in rodents was reported to be modulated by task context[28]. Given the fact that the OFCs in rodents and primates are not evolutionarily homologues, such a discrepancy may highlight an interesting difference previously unknown between the two species.

Finally, our results suggest that the representations of spatial attention and reward expectancy are dissociable in the brain. The modulation effects of attention and reward on neuronal activities are similar, and many previous studies have not clearly distinguished attention and reward in the experiment design[2]. Here, we demonstrated dissociable representations of reward salience and spatial attention in the prefrontal cortex. Spatial attention was well represented only in the DLPFC. The reward salience signal remains stable in both the OFC and the DLPFC even when spatial attention shifted sides via the top-down mechanism. Such a representation of the salience signal independent of top-down control may be necessary for the brain to evaluate the potential targets toward which spatial attention could be directed. Future experiments could be designed to investigate this hypothesis.

## Methods
### Subjects
Two male rhesus monkeys (*Macaca mulatta*) were used. They weighed 9.3 kg (subject D) and 6.8 kg (subject G) at the beginning of the training. All procedures followed the protocol approved by the Animal Care Committee of Shanghai Institutes for Biological Sciences, Chinese Academy of Sciences (CEBSIT-2021004).

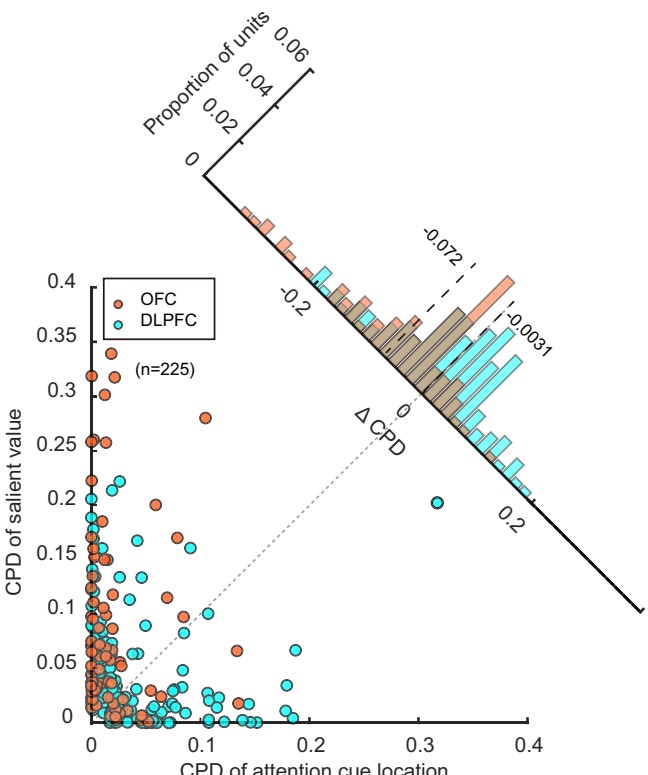

**Fig. 7 | Comparison of the spatial attention and value signals between OFC and DLPFC neurons.** The CPDs of the attention cue location are plotted against the CPDs of SV for individual OFC (orange) and DLPFC (magenta) neurons. The dotted line is the diagonal. Only the neurons with non-zero coefficients for the attention cue location or SV were plotted ($n = 225$, $p < 0.01$, one-sample $t$-test without multiple comparisons). The insert shows the distribution of the CPD differences between the two CPDs. Vertical dashed lines indicate the mean.

## Behavioral task and materials

Head-restrained monkeys were seated in a primate chair facing a 23.6 inch computer monitor at 60 cm away from their eyes. The center of the screen was adjusted to align to the midpoint of the two eyes. Behavioral tasks were run with the MATLAB based software MonkeyLogic[29]. Subjects' eye position and pupil dilation were tracked with an infrared oculomotor system at a sampling rate of 500 Hz (EyeLink 1000). Juice was delivered by a computer-controlled solenoid.

We trained two monkeys to perform a visual detection task (Fig. 1a). The monkeys had to hold their gaze at a central fixation point on the screen within a 3° wide window. After maintaining fixation for 0.5 s, a yellow saccade target was presented 6.0° (monkey D) or 7.6° (monkey G) above the fixation point. The saccade target remained on the screen until the end of the trial. After 0.5 s, a square frame (attention cue) 2.4° wide was presented on the left or right (monkey D: 7.0°; monkey G: 7.6°) of the fixation point. After another 0.2 s, two visual stimuli of 2.4° size appeared on the left and right sides (monkey D: 7.0°; monkey G: 7.6°) of the fixation point. The stimulus locations were rotated slightly (23° counter-clockwise) for monkey G to minimize its left-right bias. The attention cue disappeared 0.2 s after the stimulus onset. After a random variable delay, one of the two stimuli would change its luminance.

For monkey D, there were 90% of trials (valid-cue trials) the luminance change was on the opposite side of the attention cue (cued location). In the remaining trials (invalid-cue trials), the change was on the same side of the cue (un-cued location). In both cases, the monkey was required to report the change within 0.1–0.4 s by making a saccade

toward the eye movement target to receive a juice reward. The onset latency of the luminance change was subject to an exponential decay function (tau = 2.5 s, cut off at 1.4 s) plus 0.4 s.

To encourage monkey G to direct the attention appropriately according to the cue, it was trained to report only the luminance change at the cued location (target) but ignore the change at the un-cued location (distractor). In 80% of trials, the luminance change occurred at the cued location. Among the rest of 20% of trials, half of them had a luminance change at the un-cued location only (distractor-only trials, 10%), and the other half had first a luminance change at the un-cued location and then another at the cued location (distractor + target trials, 10%). The latencies of the luminance change at the target and the distractor locations were subject to their respective exponential decay functions (target: tau = 2.5 s, cut off at 1.9 s; distractor: tau = 1.25 s, cut off at 1.9 s) plus 0.4 s. The monkey was rewarded when they made a timely response to the target in both the target and the distractor+target trials, and when they held their fixation till the end of the trial in the distractor-only trials.

The visual stimuli were associated with different amounts of juice that the monkeys might get for a correct response. There were five stimuli for each monkey, each was associated with 1 small drop (0.033 ml), 1, 2, 4, and 8 standard drops (0.10, 0.16, 0.29, and 0.55 ml) of juice, respectively. For convenience, we label the stimulus with a small drop of juice as -0 standard drops in the figures and use 0 in the regression models. The stimuli in each trial were randomly selected from the stimulus set, and their locations (left or right) were counterbalanced. For correct responses, one of the stimuli was randomly selected, and the monkey would receive the reward associated with that stimulus. Therefore, neither the cue nor the reward salience provided information on which stimulus's associated reward would be given.

During the initial training, only one stimulus was presented on the screen, and the monkeys were rewarded with its associated juice amount if they detected the luminance change correctly. The attention cue always appeared on the opposite side of the fixation point (always valid). During the recording sessions, the single-stimulus and double-stimulus trials were interleaved. There were 7.1% and 10.1% of single-stimulus trials for monkey D and monkey G, and the rest were double-stimuli trials.

## Surgery and MRI

Before the behavioral training, both monkeys received a chronic implant of a titanium headpost. After a 2-month recovery, they were trained to perform the behavioral task until they achieved satisfactory performance. Then, the monkeys received structural Magnetic Resonance Imaging (MRI) scans for us to determine the recording chamber location. After the chamber implant surgery, a manganese-enhanced MRI scan was conducted to verify the chamber placement. The scans were carried out in a Siemens 3 T scanner.

During the surgery, the monkeys were sedated with ketamine hydrochloride (10 mg/kg), and anesthesia was then induced and maintained with isoflurane gas (1.5–2%, to effect). Body temperature, heart rate, blood-oxygen concentration, and expired $CO_2$ were monitored throughout the surgical procedures.

## Neuronal recordings

A 2 cm * 1.5 cm chamber was implanted on the surface of the left (subject G) and the right (subject D) prefrontal cortex, centered 31.5 (subject G) and 27.5 (subject D) mm anterior to the interaural line (Supplementary Fig. 1). We recorded extracellular single-unit activities with tungsten microelectrodes (FHC: 0.3–2 MΩ; AlphaOmega: 0.5–3 MΩ). Each electrode was driven by an independent microdrive (AlphaOmega EPS) through a stainless-steel guide tube. The guide tube was placed within a grid with holes 1 mm apart. The depth of the penetration was confirmed by the transitions between gray and

white matter. At most four electrodes were lowered at a time. Neuronal signals were recorded with an AlphaOmega SnR system at a sampling rate of 44 kHz. We recorded neurons from the OFC and the DLPFC. The OFC recording locations were between the lateral and medial orbital sulci in Walker's areas 11 and 13. The DLPFC neurons were recorded from the area rostral to the arcuate sulcus (8a and 8b), including both the dorsal and the ventral bank of the principal sulcus (46d and 46 v).

There were 171 recording sessions overall, 71 from monkey D and 100 from monkey G. Each recording session contained on average 1381 trials (monkey D: 1668 trials; monkey G: 1177 trials), of which 885 were correct trials (monkey D: 905 trials; monkey G: 871 trials). Neurons with <100 valid-cue (target) trials were excluded from the analyses.

Voltage signals of putative single neurons were isolated offline manually with Plexon Offline Sorter (Plexon, Dallas, TX). Neurons with poor isolation or with a lower than 1 Hz response rate were excluded. There were no additional selection criteria for neurons.

### Behavioral analyses
All the behavior analyses were based on the monkeys' performance during the recording sessions.

### Accuracy and reaction time
For monkey D, an eye movement to the saccade target within 0.1–0.4 s after the luminance change is considered a hit. Otherwise, it would be considered as a miss. For monkey G, hits and misses are similarly defined, except that any eye movements to the saccade target within the 0.1–0.4 s time window on the distractor's luminance change are counted as false alarms. The RT is defined as the time from the luminance change to the eye-movement initiation in the hit and false alarm trials.

### Neuronal selectivity
To determine how the value information was encoded by the neurons, linear regressions (*fitlm* function in Matlab Statistical Toolbox) were performed for each task variable for each individual unit in three task epochs:

$$R_n(r, t) = \beta_{0,n}(t) + \beta_{1,n}(t)*\text{variable}(r) + \varepsilon, \tag{1}$$

where $R_n(r, t)$ was the average neural response of unit $n$ for a given trial $r$ within time window $t$. In the single-stimulus trials, the only relevant task variable is stimulus value ($V_{\text{sin}}$). In the double-stimuli trials, the task variables include left stimulus value (LV), right stimulus value (RV), difference between LV and RV (LV-RV), CV, UCV, difference between CV and UCV (CV-UCV), SV, NSV, difference between SV and NSV (SV-NSV), total value (TV) and attention cue's location (att cue loc). $\varepsilon$ was independent Gaussian noise. The task epochs were the cue-stimulus epoch (0.0–0.2 s after the stimulus onset), the early stimulus epoch (0.2–0.6 s after the stimulus onset), and the late stimulus epoch (0.0–0.4 s before the luminance change). A neuron is selective to a variable if $\beta_{1,n}$ is significantly different than 0 ($p < 0.005$).

A neuron is considered as value selective if it is selective to any value variables above during any task epochs (cue-stimulus, early stimulus, and late stimulus). We further divide value selective neurons into positively tuned and negatively tuned neurons. Neurons with contradictory tuning are not included in Figs. 3, 5, and Supplementary Table 1 (e.g., positively tuned to one value variable but negatively tuned to another value variable or positively tuned during one epoch but negatively tuned during another epoch).

### Decoding spatial attention
Linear discriminant analyses (LDA, *classify* function in Matlab Statistical Toolbox) were used to decode spatial attention location from the neuronal responses of DLPFC neurons. LDA aims to classify an observation into one of the K classes through modeling the posterior probability p($B = b|A = a$), where $A$ is the observation (neural responses), and $B$ is the predictor (left attention cue or right attention cue). The posterior probability is calculated through Bayes' theorem which requires prior probability $p(B)$ and conditional probability density functions p($A = a|B = b$). LDA assumes that the probability density function given each condition is a normal distribution. Because the left attention cue and the right attention cue were equally likely, the prior was set to be 0.5.

Before conducting the LDA, we *z*-scored the responses of each neuron across all the trials and all time points (−400–600 ms around stimulus onset and −400–400 ms around luminance change). The normalized responses were further smoothed in 100 ms windows with a Gaussian kernel (sigma = 50 ms). The LDA was performed using a sliding window of 25 ms with 10 ms steps.

We constructed pseudo-neuron ensembles as follows. To balance the contribution from each neuron, we first selected the neurons with more than 100 trials in both the left attention cue condition and the right attention cue condition. We randomly chose 100 trials without replacements from each condition for each neuron and constructed the confusion matrix $X \in \mathbb{R}^{M \times T \times N}$ with the sampled trials, where $M$ is the number of trials ($M = 200$, 100 trials in each cue condition), $T$ is the number of time bins, and $N$ is the number of neurons (subject D: $N = 102$; subject G: $N = 93$). To reduce noise, we ran the principal component analysis (PCA) and kept the first $P$ components that captured at least 70% of the variance for the LDA.

To estimate how spatial attention signals fluctuated with time (Fig. 2c, e), PCAs were run on the neuron dimension of $X$ at each time bin. Only the first $P$ principal components that captured at least 70% of the total variance were used to reconstruct the subspace ($Y \in \mathbb{R}^{M \times T \times P}$). The decoder was trained and tested with the reconstructed subspace at each time bin. The posterior probability of attention cue location given the neuronal responses was used to quantify the decoder performance. The results were based on 200 independent leave-one-out cross-validations. Significance was established by two-sided paired *t*-tests comparing the actual data against the posterior probabilities calculated in the same procedure but with the shuffled data.

In Fig. 2b, d, a PCA was run on the neuron dimension of the confusion matrix that combined all time bins ($X' \in \mathbb{R}^{MT \times N}$). This ensured that the neural representation at each time bin shared the same subspace after PCA. Only the first $Q$ principal components that captured at least 70% of the total variance were used to reconstruct the subspace ($Y' \in \mathbb{R}^{MT \times Q}$). $Y'$ was reshaped into a $\mathbb{R}^{M \times T \times Q}$ matrix to enable cross-temporal decoding. The decoder was trained with the neural responses in one time bin (Fig. 2b: 25 ms time bin, stepped by 10 ms; Fig. 2d: 50–200 ms before stimulus onset), and tested with the responses in all time bins.

### Linear regression models
In Figs. 4a, d and 6a, d, linear regression models were fit to average firing rates calculated with a 25 ms time window, aligned to the stimulus onset and the luminance change, respectively. We constructed two linear regression models:

$$R_n(r, t) = \beta_{0,n}(t) + \beta_{1,n}(t)\text{attention cue loc}(r) + \beta_{2,n}(t)\text{SV}(r) + \beta_{3,n}(t)\text{NSV}(r) + \varepsilon, \tag{2}$$

$$R_n(r,t) = \beta_{0,n}(t) + \beta_{1,n}(t)\text{attention cue loc}(r) + \beta_{2,n}(t)\text{CV}(r) + \beta_{3,n}(t)\text{UCV}(r) + \varepsilon, \tag{3}$$

where $R_n(r,t)$ is the average firing rate of neuron $n$ in trial $r$ at time $t$, and the predictors were attention cue location (coded as −1 or 1), SV (0, 1, 2, 4, or 8), NSV (0, 1, 2, 4, or 8), CV (0, 1, 2, 4, or 8), UCV (0, 1, 2, 4, or 8).

Each predictor was *z*-scored. The CPD was calculated as:

$$\text{CPD}_{i,n}(t) = \frac{\text{SSE}_{-i,n}(t) - \text{SSE}_{all,n}(t)}{\text{SSE}_{-i,n}(t)}, \quad (4)$$

where $\text{CPD}_{i,n}(t)$ is the CPD of variable $i$ in neuron $n$, $\text{SSE}_{-i,n}(t)$ is the residual sum of squares of the regression model without variable $i$, $\text{SSE}_{all,n}(t)$ is the residual sum of squares of the full model. The significance of $\text{CPD}_{i,n}(t)$ is tested against the baseline (paired *t*-test, $p < 0.005$ with FDR correction for multiple comparisons), which is the CPD calculated with the firing rate in a 200 ms time window before the attention cue onset averaged across different regressors:

$$\text{CPD}_{baseline,n} = \frac{1}{3} \sum_{i=1}^{3} \text{CPD}_{i,n}(t_0), \quad (5)$$

The CPDs in Figs. 4b, d, e, f and 6b, d, e, f were calculated similarly but with the average firing rate between the stimulus onset and the luminance change. To specifically study the attention from the top-down source, CPDs in Fig. 7 were calculated with the average firing rate between the frame cue offset and the luminance change.

### Individual monkey analyses
All results presented in the main text are based on the combined data from both monkeys. Analyses based on each monkey individually can be found in Supplementary Figs. 2, 6, 8, 10, 13, 15 corresponding to Figs. 2–7 in the main text. The results are consistent.

### Reporting summary
Further information on research design is available in the Nature Research Reporting Summary linked to this article.

## Data availability
The data used in this study are available at https://doi.org/10.5281/zenodo.7090240.

## Code availability
The custom codes supporting the findings of this study are available at https://github.com/tmyang-lab/reward_salience_in_OFC.

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

## Acknowledgements
We thank John Maunsell for his comments during the preparation of the paper. We also thank Ruixin Su, Wei Kong, and Lu Yu for their help in all phases of the study. This work was supported by the National Science and Technology Innovation 2030 Major Program (Grant No. 2021ZD0203701), National Key R&D Program of China (Grant No. 2019YFA0709504), Shanghai Municipal Science and Technology Major Project (Grant No. 2018SHZDZX05), and the Strategic Priority Research Program of Chinese Academy of Science (Grant No. XDB32070100).

## Author contributions

T.Y. conceived the original idea of the study. W.Z. and Y.X. collected the behavioral and neurophysiological data and performed the analysis. W.Z. and T.Y. wrote the paper. All authors designed the experiments, discussed the results and provided the feedback on the paper.

## Competing interests

The authors declare no competing interests.

## Additional information

**Correspondence and requests** for materials should be addressed to Tianming Yang.

