## [Peer Review File · Nature Communications]

Reward Saliency but not Spatial Attention Dominates the Value Representation in the Orbitofrontal CortexREVIEWER COMMENTS

Reviewer #1 (Remarks to the Author):

Zhang et al describe behavior of two monkeys, and neural activity in both LPFC and OFC, performing an attentional task. The monkeys perform well the task by paying attention to the cued locations and by being faster in valid trials. Consistent with previous results, LPFC activity is strongly modulated by attention. However, surprisingly, OFC single-neuron activity is very weakly modulated by attention. Further, OFC activity seems only to reflect the value of the most salient (valuable) stimulus, while it ignores the less salient one. The experiment is well designed, and the conclusions are supported by careful data analysis and interpretation. The results challenge current conceptions of OFC value encoding.

I would like to propose some additional analysis to further validate some of the surprising results:

-Line 236: the authors can repeat the decoding analysis by using correct and incorrect trials instead of fast and slow RTs. If the authors are correct, then in correct trials the cued location should be better decoded than in incorrect trials.

-It would be very interesting to see the response of neurons in 3D plots. First, it would be good to plot a few examples of neurons where firing rate is plotted vs SV and NSV simultaneously, that is, using 3 coordinates in total. The response surface should be flat in the NSV axis. Similarly, a few neurons could be shown where the firing rate is plotted vs CV and UCV. The surface should be quite symmetrical around the diagonal. If single neurons are too noisy, an average for positively and for negatively tuned neurons could be used instead.

-If the authors are correct, then they can use decoders based on pseudo populations to predict CV and UCV and show that CV is much better predicted than UCV.

-The sentence in line 64 can be further supported by the reference

A neuronal theory of sequential economic choice - Benjamin Y. Hayden, Rubén Moreno-Bote, 2018 (sagepub.com)

<https://journals.sagepub.com/doi/full/10.1177/2398212818766675>

-Previous work on rat OFC has shown that task engagement strongly modulated neuronal activity of single neurons

Lateral orbitofrontal cortex anticipates choices and integrates prior with current information | Nature Communications

<https://www.nature.com/articles/ncomms14823>

somehow opposite to the results shown here that attention very weakly modulate value encoding. A comparison between the two set of results would be of high value.

Minor

-Line 323: Typo: sat -> saturated?

-In panel 4b and 4e, use scientific notation for the p values

Reviewer #2 (Remarks to the Author):

The manuscript by Zhang, Xie and Yang investigates the representation of reward and spatial attention in two regions of the frontal cortex: the lateral prefrontal cortex (LPFC) and the orbitofrontal cortex (OFC). The authors use a beautiful paradigm with cues that signal a cue for the required timing of a stereotypical eye movement and additional shape cues that signal the amount of reward that can be expected. The results are interesting and they also seem robust although there were some unexpected differences between the tasks of the two monkeys. I have several recommendations regarding the terminology, theoretical frameworks that include attention and reward expectancy and the comparison of results between the two brain regions. Some of my concerns stand in the way of recommending that this paper should be published, but I do expect that these concerns might be addressed by additional analysis and, in particular, a much more careful framing of the roles of attention and reward.

Major concerns:

1) The present paradigm has several aspects that differ from previous paradigms in interesting ways, although there are also a few troublesome issues. The most interesting trials are those in which two shapes are presented, which both cue a possible reward amount. This design dissociates the cue location from the expected reward. Thus, even though both cues signal reward their precise location is irrelevant. There also is an attention cue, which signals the most likely place where a luminance go-cue will appear. In one monkey the cue indicates the location of the most likely luminance change, but the relevant luminance change may also happen at the uncued location. In the other monkey, the luminance change on the uncued side should be ignored. There are a few issues with this approach:

- It is an error (even misleading) to pool the invalidly cued trials of one monkey with the false alarm trials of another monkey and pretend that they convey the same information. The authors should inform the reader in the main text about this difference in the design between the monkeys and why this approach was chosen. Was the desired behavioral effect not obtained with one of the paradigms in one of the monkeys? It is more logical to use the same paradigm for both monkeys. Figures that erroneously pool data from non-comparable paradigms using incompatible measures should be removed from the Ms. This includes all panels of Fig. 1, which should be replaced by a proper analysis. If that analysis needs to differ between the two monkeys, then that is the way it is.
- Related to this, in line 119 the authors state "After the training, the monkeys performed the task well. When the cue was valid, they were more accurate (valid: 93.0%, SEM=0.2%; invalid: 84.7%, SEM=0.6%. $p < 0.001$, two-tailed Wilcoxon signed rank test)". However, in one of the monkeys the cue could not be invalid. This is a misrepresentation of the results. By the way, what is the meaning of this SEM? Across monkeys? Across trials? Across sessions?
- Related: in line 590 the authors mention a "target stimulus". For one monkey either shape cue could change luminance right? So for this monkey there was no target stimulus?
- What was the proportion of trials in which the monkey erroneously responded to the luminance change that he/she was trained to ignore? Did the other shape change luminance later in the same trial or was the monkey rewarded for maintaining fixation?
- Was the luminance change sometimes missed? How often?
- How many recording sessions were done with the two monkeys? How many trials, on average, per session? Did the authors keep recording from the same cells during a session, or were new cells sought after a predetermined number of trials? What was the percentage of trials with one shape cue and with two shape cues?
- Fig. 1b: why are there only two red bars "salient" vs. "non-salient" (that should read high and low reward, see below, point 3) while there were five levels of reward magnitude?
- The authors compare the influence of the spatial attention cue to the influence of reward magnitude (lines 146-156) on "performance", i.e. reaction time and accuracy. For the reward magnitude effect, they compare the trials in which the high-reward shape and the low-reward shape were cued and changed luminance. However, the spatial location of this cue (left vs. right) did not have any relevance for reward expectancy. Consequently, the weak effect of reward magnitude on the accuracy in this paradigm might be the result of this idiosyncratic feature of the task design, which is very different from many natural, e.g. foraging situations. I therefore disagree with the statement in line

415 "The results strongly argue against the previously proposed theory that attention serves as a selection mechanism for the OFC's value encoding". That may be true for the present paradigm but must be different in other paradigms where reward expectancy only be estimated after attending particular items.

- For the analysis where this can be done: please tell the reader that results were replicated across the two individual monkeys, mentioning the relevant supp info figure at the appropriate location in the Ms. I was concerned about interindividual differences when reading the Ms and was only later reassured when opening the Supp Info.

2) I am concerned about the unbalanced approach of analyzing data from LPFC and OFC. I'd like to know if there are any LPFC-like cells in OFC and vice versa, or was tuning completely different? Now the authors start with a selection of neurons that have a desired property: tuning of the attention cue location for LPFC and tuning to reward magnitude in OFC and their entire analysis is based on these selections. With this approach the two brain regions can only seem different and a direct comparison of the two brain regions using the same measures is lacking. One therefore wonders if there are any cells in OFC that are tuned to the attention cue location and if there are any cells in LPFC tuned to reward magnitude. My recommendation is to add a direct comparison between these brain areas early in the Ms and even replicate the analyses (where possible) across brain regions.

- A possible outcome is that this does not make sense because the neurons simply do not have the required selectivity (e.g. OFC neurons do not care about spatial attention and LPFC neurons do not care about reward magnitude). If that is the case, it should first be quantified and also be reported.

- To follow up on this point, I find it very unlikely that LPFC neurons do not care about reward magnitude because such signals have even been found in V1 (Stanisor et al. 2013, PNAS).

- Vice versa, there does seem to be spatial cue decoding in OFC too (e.g. Fig. 4).

3) I am concerned with the terminology which is in many places confusing.

- The use of the word "saliency" for reward value is unnecessary and confusing. Saliency is used by different authors in different manners, causing a lot of confusion in the literature. For example, some use it for the stimulus-driven "attention-grabbing" ability of stimuli, others for task relevance, and here it is used for reward value. The authors should simply replace this term by "reward value" to avoid this confusion and thereby make the text more comprehensible. There is no need to use saliency here.

- Same with "salient value" and "non-salient value". Replace this by "higher value" and "lower value", which is the more precise terminology.

- "Cued value" and "non-cued value" are also misnomers. The monkey is cued to attend an item that will change luminance. But this attention cue has no impact on the value, hence it does not make sense to talk about "cued value". It is the shape of the two cues (and not their luminance nor their locations) that jointly determine the expected value, because that is the average. Hence, the attention cue has no predictive value for the amount of reward that the monkey can expect to receive.

- The location cue is suggested to cue "top-down attention" and the shape cues "bottom-up" reward saliency. Why is one top-down and the other bottom-up? The spatial cue tells the monkey the most likely location of the luminance change that instructs the timing of an eye movement response. I.e. it is an instruction to attend a spatial location. The shape cues, in their combination, inform the monkey about the amount of the reward that the animal should expect after the trial. The information about both cues is presumably held in working memory for a while. The location cue needs to be in working memory because the monkey needs to remember where the luminance change will occur that is the go-cue for the eye movement response. The shape cue is in working memory because the monkey prepares for the amount of reward it will receive. Why would one cue be bottom-up and the other top-down as they are both in working memory? What is the logic here? I believe that both cues should be treated on equal footing and that the term "top-down" should be removed (everywhere).

- It might be useful to rewrite the section that includes a reference to Maunsell (2004 TiCS) who actually discussed that it is difficult, if not impossible, to dissociate reward from attention in NHP work. Indeed, in the present paradigm the spatial cue (frame) is important to receive reward (it is thereby also a sort of reward cue) because if the eye movement instructed by the luminance change at the

cued location is too late or too early no reward will be obtained. The authors might want to revise their statement in line 463 that attention and reward can be dissociated in the brain, for two reasons. First, this statement is about spatial cues and shape cues, which in the present paradigm instruct or give information about different aspects of the task and are all necessary to get the reward at the end. The outcome would have been different with another design of the task. Second, if the dissociation is about the different roles of DPFV vs. OFC, I noted in my point 2 that this dissociation was not achieved convincingly.

- To follow up on the last point, it seems that the authors' reasoning implies that there is only one type of attention and that if attention facilitates processing at a particular spatial location, then that is where attention is. However, there are many forms of attention, which can also be object based, feature based, etc. that all be expressed at the same time. This is another reason to keep the terminology closer to the experimental design and why I advise to use terms like "reward magnitude cue" and "spatial cue" rather than "top-down attention" and "bottom-up value". One example statement where this is an issue is in line 418 "The dominance of reward saliency that we observed cannot be attributed to top-down attention, decision making, or eye movements". That statement holds in the present paradigm for spatial attention to the most likely luminance change location, but it may be wrong for object-based attention and feature-based attention, which may be directed to shapes, in particular shapes that predict a high reward value. We know from previous work by the lab of e.g. Desimone that there are different circuits and neurons coding for shape and feature based attention. The same problem occurs in line 461 suggestion that top-down attention cannot help very much. Help with what?

4) The response of the OFC neurons was mainly driven by the stimulus with highest reward value and the value of the stimulus associated with less reward had little influence on activity (replicating an earlier study by the same team, Xie et al., 2018). One wonders whether this insensitivity to the lower reward value was also expressed in behavior. For example, did the reaction time or the probability that the monkeys aborted a trial (e.g. broke fixation) also not depend on the lower reward value? - Did the authors more generally look at the probability of fixation breaks and the speed of the saccade as markers of reward expectancy (see e.g. Takikawa et al. (2002) Modulation of saccadic eye movements by predicted reward outcome. *Exp Brain Res* 142:284–291)?

5) The quality of the figures was not all that good, but I noticed that the cell of Fig. 3a start coding for reward expectancy before the stimuli appear on the screen. Is that real? I am curious because the same effect occurs in Fig. 3b. Or is that a smoothing effect? Please explain.

6) It is odd that 0 drops in the paper actually correspond to 0.033ml (line 609). Hence 0 drops is also a bit misleading, why not simply putting reward magnitude in ml, to accurately describe what was done?

7) It would be good to explain the regression analysis in lines 663 and following a bit better. If CV and UCV are predictors then adding CV-UCV as an extra predictor cannot work, because there is no variance that remains to be explained by the difference. Can the authors spell out how this was done?

8) Supplementary Figs. 5 and 6 were missing from the Ms, and I have therefore not been able to judge the individual monkey results of these analyses.

Smaller points:

Please make it very clear at the beginning of the results section that the location of the two shape cues does not convey any information about reward expectancy. It took me some time to realize that this is the case.

Line 77: "The monkeys had to direct their attention [...] to detect the luminance change". Really? How

do the authors know that they could not detect the change without attention. This statement is erroneous because one monkey should completely ignore the luminance change at the non-cued location.

Line 86: What is "an early stage of reward processing"? Do the shape not have to be processed from early visual cortex up to the category level and then in OFC? If so, what is "early" about it?

In the discussion it would be useful to include a few lines explaining how the present results relate and go beyond Xie et al. 2018.

Trials in which a high reward is at stake might cause a state of general attentiveness or arousal (19). Furthermore, reward value might influence "selective" attention; in the presence of multiple stimuli, attention might be attracted to those that are more rewarding.

Studies on the neuronal correlates of expected value may therefore have measured attention shifts (29, 32).

Typo on line 431: comes -> come

Line 436: I disagree that the frame was "valueless" because it noted progress of the monkey through the trial and indicated that fixation has been attained for some time, that the trial had started and hence that a new reward was soon to be obtained. Same line: "bottom-up signal": not sure it was also registered in working memory and therefore would also have top-down properties.

Reviewer #3 (Remarks to the Author):

It is well established that neural activity in the orbitofrontal cortex tracks the value of predictive stimuli. The current experiment tests whether this value coding is controlled by top-down attentional mechanisms or by the salience (learned value) of stimuli. Two monkeys were trained on a Posner-like task in which they had to respond to a luminance change in either of two stimuli presented on the left- and right-hand side of the screen. The side with the luminance change was predicted by an attentional cue (80 or 90% validity). The two stimuli were previously associated with different amounts of reward. Upon a correct response, either of the two reward magnitudes predicted by the stimuli were paid out (50% chance for either). The "salient stimulus" was defined as the stimulus with the higher value. Behavior (accuracy and response times (RT)) was affected by both the location of the attentional cue and the location of the salient stimulus. Activity of LPFC neurons tracked animals' spatial attention, whereas activity of OFC neurons tracked the value of the salient stimulus, regardless of whether it was cued or non-cued, though there was a slight preference to encode the value of the cued stimulus.

The manuscript has several strengths. The question of what determines value encoding in the OFC is important and timely, and the neural recordings from LPFC to track animal's attention is very nice. However, I have several major conceptual and methodological concerns that dampen my enthusiasm. Most importantly, I don't think the conclusion that "the results strongly argue against the previously proposed theory that attention serves as a selection mechanism for the OFC's value encoding" is justified.

Major concerns

1. My most important concern is that I don't think that the behavioral task can provide meaningful insights into the mechanisms controlling value encoding in the OFC.

1.1. The reward delivered on a given trial was randomly selected from the two presented stimuli. But their value was otherwise irrelevant for guiding behavior. The only thing that mattered to obtain the reward was to detect and respond to the luminance change. Thus, the cue likely directed attention to the luminance of the stimuli not their shape. Based on this task design, it is unsurprising that attentional cues did not strongly affect value signals in OFC.

1.2. There is also no benefit of selectively paying more attention to the value of any of the two stimuli. Rather, the only potentially relevant value was the average value predicted by the two stimuli, as this signals the amount of reward the animal can expect. But whether OFC activity tracked the average value was not tested. And even if OFC codes the higher value of the stimuli, this could simply reflect an optimism bias.

1.3. Because the current task does not allow animals to engage in value-based decision-making of any sort, it is unclear why this would be a good way to study the role of OFC in value-based decision-making. There is no deliberation between different options like in the task used by Rich and Wallis, where animals could make choices between alternative options. There is simply no reason to do this in the current task where the expected reward on a given trial was independent of which stimulus was attended.

2. I am also not convinced that the main conclusions are well supported by the data.

2.1. Although more neurons tracked the value of the salient stimulus ($105/357 = 30\%$), a large number of OFC neurons ($34/357 = 10\%$) did track the value of the non-salient stimulus. This could suggest that different populations of OFC neurons track the value of the different stimuli. However, this is contrary to the authors' conclusion that OFC ignored the value of the non-salient stimulus.

2.2. Similarly, although OFC neurons tracked both the value of the cued and non-cued stimulus, at the individual neuron level, there was a preference for the cued over the non-cued values. Again, this is contrary to the main conclusion that top-down attention does not affect value coding in the OFC.

3. The experiment is fundamentally a two factorial design (salient vs. non-salient X cued vs. non-cued). Throughout the manuscript, the authors report the averages for each factor level across the other factor, rather than reporting the four individual cells separately. This would be necessary to see whether there are any interactions between the two attentional mechanisms.

3.1. Please add an analysis similar to Figure 3j,k,i, for all four conditions separately (salient-cued, non-salient-cued, salient-non-cued, non-salient-non-cued)

3.2. Again, related to Figure 4, it would be important to show whether neurons track the salient and non-salient value separately for cued and non-cued locations. That is, please compute CPD for all four conditions (salient-cued, salient-non-cued, non-salient-cued, and non-salient-non-cued). This could show whether, for instance, the non-salient value is more strongly encoded when it is cued.

3.3. Finally, it would be important to show and analyze behavior (Figure 1b-e) based on each of the four possible conditions (valid-salient, valid-non-salient, invalid-salient, invalid-non-salient) rather than the averages for valid/invalid and salient/non-salient.

4. It is unclear what constitutes an incorrect response in this task. There was only one response target, and so animals had to simply indicate that any of the two stimuli had changed their luminance. This would mean that any response made on any trial was correct. So what was an incorrect response? If the monkey did not respond? Or if they responded too early or too late?

Please find our reply to the reviewers. We are grateful for many valuable suggestions, which we have incorporated in our revision. The most notable changes in the revised paper are as follows.

1. We have made extensive changes in Results and Discussion to reflect reviewers' comments on top-down vs. bottom-up attention. While we continue to believe reward salience comes from a bottom-up source, we avoid using bottom-up to describe the results and only state our opinion in Discussion. Similarly, we have refrained from using the word "top-down" too much, except when we are specifically referring to the attention that is directed by the cue. In other places, we use the term "spatial attention" instead.
2. We now illustrate monkeys' behavior performances separately to reflect their task paradigms' differences. The results indicated that both monkeys allocated their spatial attention according to the task demand.
3. We have provided parallel analyses of the OFC and the DLPFC data. The results were consistent with our conclusion. We found that the value encoding was modulated similarly by reward salience in the OFC and the DLPFC, while the spatial attention signals were only encoded in the DLPFC.
4. We have provided further analyses based on the reviewers' suggestions. The results provide further support to our conclusions.

We hope the reviewers agree that our revision is a better paper with all these changes. Below the referees' comments are quoted in black; our responses are in blue.

Reviewer #1 (Remarks to the Author):

Zhang et al describe behavior of two monkeys, and neural activity in both LPFC and OFC, performing an attentional task. The monkeys perform well the task by paying attention to the cued locations and by being faster in valid trials. Consistent with previous results, LPFC activity is strongly modulated by attention. However, surprisingly, OFC single-neuron activity is very weakly modulated by attention. Further, OFC activity seems only to reflect the value of the most salient (valuable) stimulus, while it ignores the less salient one. The experiment is well designed, and the conclusions are supported by careful data analysis and interpretation. The results challenge current conceptions of OFC value encoding.

I would like to propose some additional analysis to further validate some of the surprising results:

We'd like to thank the reviewer for the positive and constructive comments. We have made the proposed analyses and included them in the revision. The results are consistent with the original findings and strengthen our claim.

-Line 236: the authors can repeat the decoding analysis by using correct and incorrect trials instead of fast and slow RTs. If the authors are correct, then in correct trials the cued location should be better decoded than in incorrect trials.

Thanks for suggesting the analysis. We have performed the decoding analyses on hit and miss trials. As expected, the decoders' performance in the hit trials was slightly above that in the miss trials. The new analysis is now included as **Supplementary figure 3** and mentioned in Results.

-It would be very interesting to see the response of neurons in 3D plots. First, it would be good to plot a few examples of neurons where firing rate is plotted vs SV and NSV simultaneously, that is, using 3 coordinates in total. The response surface should be flat in the NSV axis. Similarly, a few neurons could be shown where the firing rate is plotted vs CV and UCV. The surface should be quite symmetrical around the diagonal. If single neurons are too noisy, an average for positively and for negatively tuned neurons could be used instead.

Thanks for the suggestion. We plotted the OFC population responses against the SV and the NSV and against the CV and the UCV. The results are what one would expect. In the SV/NSV plot, the neurons' responses form parallel traces by the SV, suggesting they were mainly modulated by the SV. In the CV/UCV plot, the neuronal responses were quite symmetric for the CV and the UCV, suggesting that they were modulated similarly by both the CV and the UCV. The new figures are now included as **Supplementary figures 4, 5**.

-If the authors are correct, then they can use decoders based on pseudo populations to predict CV and UCV and show that CV is much better predicted than UCV.

As we show in the answer to the above point, the neuronal responses were quite similar for the CV and the UCV (**Supplementary figures 4 and 5**). To further demonstrate this point, we decoded the CV and the UCV with OFC neuronal responses (**Figure R1**) as the reviewer suggested. We found that the decoding performance for the CV was only significantly better than that for the UCV at a few time points (black segments at the top) before the luminance change. This is consistent with **Figure 4d, e, f**, in which we show that the CV was slightly better encoded by the OFC neurons than the UCV using the CPD analyses. The cue and its associated spatial attention only weakly modulated OFC neurons' value encoding.

As **Figure R1** only provides limited extra information, we decide not to include it in the paper but would be happy to oblige if the reviewer thinks otherwise.

Figure R1. Decoding of the CV and the UCV. The posterior probability of the value at the cued location (red) and the un-cued location (purple) across trials are decoded from OFC pseudo neuronal population ensemble activities in both monkeys. Significance is assessed with two-tailed paired t-tests ($p < 0.01$ with FDR corrections for multiple comparisons). The red segments at the top indicate when the CV was significantly encoded in the population (tested against the data with shuffled CV labels). The purple segments indicate when the UCV was significantly encoded (tested against the data with shuffled UCV labels). The black segments indicate when the posterior probability of the CV and the UCV were significantly different. Thin grey lines represent SEM across trials.

-The sentence in line 64 can be further supported by the reference

A neuronal theory of sequential economic choice – Benjamin Y. Hayden, Rubén Moreno-Bote, 2018 ([sagepub.com](https://www.sagepub.com))

Thanks for the suggestion. We have included the suggested reference.

-Previous work on rat OFC has shown that task engagement strongly modulated neuronal activity of single neurons

Lateral orbitofrontal cortex anticipates choices and integrates prior with current information | Nature Communications

<https://www.nature.com/articles/ncomms14823>

somehow opposite to the results shown here that attention very weakly modulate value encoding. A comparison between the two set of results would be of high value.

Thanks for the suggestion. The current results appear to contradict some of the previous findings in the rodents' OFC. However, given the fact that the OFCs in the two species are not evolutionarily homologues, such a discrepancy may highlight the difference between the two species. We have added the following sentences in Discussion:

However, the findings of the dominance of SV encoding and the lack of top-down modulation may not generalize to other species. Notably, the OFC in rodents was reported to be modulated by task context (Nogueira et al., 2017). Given the fact that the OFCs in rodents and primates are not evolutionarily homologues, such a discrepancy may highlight an interesting difference previously unknown between the two species.

Minor

-Line 323: Typo: sat -> saturated?

We have replaced the word with *clustered*.

-In panel 4b and 4e, use scientific notation for the p values

Fixed.

Reviewer #2 (Remarks to the Author):

The manuscript by Zhang, Xie and Yang investigates the representation of reward and spatial attention in two regions of the frontal cortex: the lateral prefrontal cortex (LPFC) and the orbitofrontal cortex (OFC). The authors use a beautiful paradigm with cues that signal a cue

for the required timing of a stereotypical eye movement and additional shape cues that signal the amount of reward that can be expected. The results are interesting and they also seem robust although there were some unexpected differences between the tasks of the two monkeys. I have several recommendations regarding the terminology, theoretical frameworks that include attention and reward expectancy and the comparison of results between the two brain regions. Some of my concerns stand in the way of recommending that this paper should be published, but I do expect that these concerns might be addressed by additional analysis and, in particular, a much more careful framing of the roles of attention and reward.

Major concerns:

1) The present paradigm has several aspects that differ from previous paradigms in interesting ways, although there are also a few troublesome issues. The most interesting trials are those in which two shapes are presented, which both cue a possible reward amount. This design dissociates the cue location from the expected reward. Thus, even though both cues signal reward their precise location is irrelevant. There also is an attention cue, which signals the most likely place where a luminance go-cue will appear. In one monkey the cue indicates the location of the most likely luminance change, but the relevant luminance change may also happen at the uncued location. In the other monkey, the luminance change on the uncued side should be ignored. There are a few issues with this approach:

- It is an error (even misleading) to pool the invalidly cued trials of one monkey with the false alarm trials of another monkey and pretend that they convey the same information. The authors should inform the reader in the main text about this difference in the design between the monkeys and why this approach was chosen. Was the desired behavioral effect not obtained with one of the paradigms in one of the monkeys? It is more logical to use the same paradigm for both monkeys. Figures that erroneously pool data from non-comparable paradigms using incompatible measures should be removed from the Ms. This includes all panels of Fig. 1, which should be replaced by a proper analysis. If that analysis needs to differ between the two monkeys, then that is the way it is.

We admit that it is not ideal that two monkeys performed different tasks, yet it was critical to train the two monkeys to use attention appropriately. In particular, we found that monkey G did not show any behavior improvements when responses to the un-cued changes were allowed as in monkey D. Therefore, we forced monkey G not to report the luminance change at the un-cued location. By adding this restriction, we found that monkey G's behavior was consistent with appropriate attention assignment. Both types of attentional manipulation schemes we used for the two monkeys can be found in the attention literature^{1,2}, and the neuron analyses further supported the behavior results.

To state the reason for this task difference, we revised Results as follows:

... to encourage monkey G to assign its attention appropriately, we required it to ignore the luminance change at the un-cued location and not to make a response...

In the new revision, we explained the task difference in more detail, analyzed two monkeys' behavior separately, and revised **Figure 1** and the corresponding Results section accordingly. The conclusions remain the same. Both monkeys learned to direct their attention to the appropriate location.

- Related to this, in line 119 the authors state “After the training, the monkeys performed the task well. When the cue was valid, they were more accurate (valid: 93.0%, SEM=0.2%; invalid: 84.7%, SEM=0.6%. $p < 0.001$, two-tailed Wilcoxon signed rank test)”. However, in one of the monkeys, the cue could not be invalid. This is a misrepresentation of the results. By the way, what is the meaning of this SEM? Across monkeys? Across trials? Across sessions?

As mentioned above, we have revised the section and explained the results from the two monkeys separately.

All SEMs in **Figure 1** are across sessions, which is now stated in the paper.

- Related: in line 590 the authors mention a “target stimulus”. For one monkey either shape cue could change luminance right? So for this monkey, there was no target stimulus?

To clarify, we have revised the Results as follows.

... For monkey D, the cue was valid in 90% of the trials. In the remaining trials (invalid-cue trials), the change location was on the same side of the frame (un-cued location). Monkey G was also trained to detect the luminance change at the cued location (target trials: 80%). In addition, to encourage monkey G to assign its attention appropriately, we required it to ignore the luminance change at the un-cued location and not to make a response (distractor trials: 20%). In half of the distractor trials, another luminance change would happen at the cued location after the distractor (distractor+target trials: 10%), and monkey G was rewarded for detecting the change. In the other half of the distractor trials (distractor-only trials: 10%), the target stimulus did not change its luminance, and the monkey held fixation till the end of the trial to receive a reward.

- What was the proportion of trials in which the monkey erroneously responded to the luminance change that he/she was trained to ignore? Did the other shape change luminance later in the same trial or was the monkey rewarded for maintaining fixation?

For monkey G, there were three trial conditions: a luminance change at the cued location (target trials, 80%), a luminance change at the un-cued location only (distractor-only trials, 10%), and a luminance change first at the un-cued location and then another at the cued location (distractor+target trials, 10%). Monkey G was rewarded when it made a timely response to the luminance change at the cued location in the target and the distractor+target trials, and when it held fixation till the end in the distractor-only trials.

Any responses that occurred within 100-400 ms time window after the luminance change at the un-cued location in either distractor-only or distractor+target trials are counted as false alarms. Monkey G had an $18.6 \pm 0.4\%$ (SEM across sessions) false alarm rate.

Please refer to the new **Figure 1** for further details.

- Was the luminance change sometimes missed? How often?

For monkey D, the miss rate was 5.8% (SEM=0.2%) in the valid trials and 9.7% (SEM=0.4%) in the invalid trials. For monkey G, the miss rate was 7.8% (SEM=0.3%) in the target trials, and 10.0% (SEM=0.6%) in responses to the 2nd luminance change in the distractor+target trials (**Figure 1g, i**).

- How many recording sessions were done with the two monkeys? How many trials, on average, per session? Did the authors keep recording from the same cells during a session, or were new cells sought after a predetermined number of trials? What was the percentage of trials with one shape cue and with two shape cues?

There were 171 recording sessions overall, 71 from monkey D and 100 from monkey G. Each recording session contained on average 1381 trials (monkey D: 1668 trials; monkey G: 1177 trials), of which 885 were correct trials (monkey D: 905 trials; monkey G: 871 trials).

There was not a predetermined number of trials for recording, and we kept recording from the same cell until either the monkeys quit working, or we lost the cell. On average, the recording data for each cell contained 547 trials, of which 330 were correct. Neurons with less than 100 correct valid-cue trials in monkey D or 100 correct target trials in monkey G would not be included for analysis.

There were 7.1% and 10.1% of single-shape trials for monkey D and monkey G, and the rest were double-shape trials.

- Fig. 1b: why are there only two red bars “salient” vs. “non-salient” (that should read high and low reward, see below, point 3) while there were five levels of reward magnitude?

In the original **Figure 1b**, Salient labeled the trials in which the luminance occurred at the salient stimulus, and non-salient labeled the trials in which the luminance occurred at the non-salient stimulus. Therefore, there were not 5 levels of reward magnitude plotted separately. The new **Figure 1b,c,e,f** are similarly labeled.

- The authors compare the influence of the spatial attention cue to the influence of reward magnitude (lines 146-156) on “performance”, i.e. reaction time and accuracy. For the reward magnitude effect, they compare the trials in which the high-reward shape and the low-reward shape were cued and changed luminance. However, the spatial location of this cue (left vs. right) did not have any relevance for reward expectancy. Consequently, the weak effect of reward magnitude on the accuracy in this paradigm might be the result of this idiosyncratic feature of the task design, which is very different from many natural, e.g. foraging situations. I therefore disagree with the statement in line 415 “The results strongly argue against the previously proposed theory that attention serves as a selection mechanism for the OFC’s value encoding”. That may be true for the present paradigm but must be different in other paradigms where reward expectancy only be estimated after attending particular items.

This is exactly our point. The effect of reward magnitude on behavior was very weak. Yet, it dominated OFC’s value encoding. In contrast, the frame cue and its associated spatial attention was important for behavior but only affected OFC’s value encoding weakly. The argument would be much weaker if we used a task in which reward magnitude was important for decision.

In some task paradigms, such as in McGinty and Lupkin, 2021³, an item's reward or value is only revealed after an eye movement choice is made. In such cases, we certainly won't see any attentional modulation of value encoding before attention is assigned to the item, as value information is not available yet.

The attention we referred to in "attention serves as a selection mechanism for the OFC's value encoding" is the top-down attention associated with the cue. We are considering the case when multiple items are presented simultaneously, and their value associations are clear to the subjects. In this situation, we find that the OFC mainly encodes the value of the most rewarding item, but not the value of the attended item as previously proposed in Xie et al., 2018⁴.

To avoid confusion, we have changed the statement to:

The results argue against the previously proposed theory that attention serves as a selection mechanism for the OFC's value encoding when multiple items with different value associations are presented.

- For the analysis where this can be done: please tell the reader that results were replicated across the two individual monkeys, mentioning the relevant supp info figure at the appropriate location in the Ms. I was concerned about interindividual differences when reading the Ms and was only later reassured when opening the Supp Info.

Thanks for the suggestion. We have made the changes in Results accordingly.

2) I am concerned about the unbalanced approach of analyzing data from LPFC and OFC. I'd like to know if there are any LPFC-like cells in OFC and vice versa, or was tuning completely different? Now the authors start with a selection of neurons that have a desired property: tuning of the attention cue location for LPFC and tuning to reward magnitude in OFC and their entire analysis is based on these selections. With this approach the two brain regions can only seem different and a direct comparison of the two brain regions using the same measures is lacking. One therefore wonders if there are any cells in OFC that are tuned to the attention cue location and if there are any cells in LPFC tuned to reward magnitude. My recommendation is to add a direct comparison between these brain areas early in the Ms and even replicate the analyses (where possible) across brain regions.

- A possible outcome is that this does not make sense because the neurons simply do not have the required selectivity (e.g. OFC neurons do not care about spatial attention and LPFC neurons do not care about reward magnitude). If that is the case, it should first be quantified and also be reported.

- To follow up on this point, I find it very unlikely that LPFC neurons do not care about reward magnitude because such signals have even been found in V1 (Stanisor et al. 2013, PNAS).

- Vice versa, there does seem to be spatial cue decoding in OFC too (e.g. Fig. 4).

Our paper aims at examining how attention modulates the value encoding in the OFC. The DLPFC analyses, together with the behavior analyses, are used to demonstrate that the attention manipulation was successful. Therefore, we approached the two areas with distinct analyses.

We'd also like to point out that we did not select neurons that had a desired property during the recording sessions. We recorded and analyzed all neurons that we encountered in both areas as long as these neurons passed quality control and had enough trials (see our selection criteria in Methods, Neural recording part, paragraph 2). Only for hypothesis testing, the analyses were oriented differently for each area.

Nevertheless, as the reviewer demanded, we now include all parallel analyses for both the OFC and the DLPFC. Some are included in the main text, and the others in the supplementary. The new analyses and figures are as follows.

1. To study whether the OFC encoded cue location (top-down spatial attention), we did the same analyses for the OFC as we did for the DLPFC in **Figure 2** and showed the results in **Supplementary figures 11**. Unlike the DLPFC, the OFC did not encode the top-down spatial attention after the cue offset. This is consistent with the previous studies that showed the OFC lacks spatial sensitivity.
2. To demonstrate the value encoding of DLPFC neurons, we did the same analyses for the DLPFC as we did for the OFC in **Figure 3**, and the results are shown in **Supplementary figure 12**. For simplicity, we did not include the PSTH (as in **Figure 3a-f**) in this plot. Overall, we found that value encoding in the DLPFC was also dominated by reward salience, although there was a strong spatial attention signal in the DLPFC.
 - a. In **Supplementary figure 12a, c, e, g, i, k**, we assessed whether the DLPFC responses could be explained by the SV. We compared their responses to two identical stimuli against the responses to pairs of different stimuli (**Supplementary figure 12a, c**). We found that when sorted by the SV, the responses to the different-stimuli pairs were not significantly different from the responses to the same-stimuli pairs for the positively tuned neurons (group: $F_{1,674}=0.45$, $p=0.50$; value: $F_{4,674}=38.05$, $p<<0.001$ in two-way ANOVA). The negatively tuned neurons had slightly but significantly higher responses to the different-stimuli pairs (group: $F_{1,724}=4.24$, $p=0.04$; value: $F_{4,724}=77.58$, $p<<0.001$ in two-way ANOVA), indicating they were affected slightly by the NSV.
 - b. Similarly, we also compared the DLPFC neuronal responses grouped by the CV and the UCV (**Supplementary figure 12b, d**) as we did for the OFC neurons in **Figure 3j, k, l**. We did not find any differences between them in either the positively tuned neurons (group: $F_{1,674}=0.02$, $p=0.89$; value: $F_{4,674}=52.96$, $p<<0.001$ in two-way ANOVA) or the negatively tuned neurons (group: $F_{1,724}=0.06$, $p=0.80$; value: $F_{4,724}=112.17$, $p<<0.001$ in two-way ANOVA).
3. To quantitatively assess DLPFC's encoding of the SV/NSV and the CV/UCV, we did the same analyses for the DLPFC as we did for the OFC in **Figure 4** and added the new **Figures 6** in the main text to show the results.
 - a. We assessed how well the DLPFC responses could be explained by the SV and the NSV. The average CPDs of the SV were significantly above the baseline right after the stimulus onset and lasted until the end of the trials. The CPDs of the SV were much larger than the CPDs of the NSV during the whole stimulus period (**Figure 6a**). The difference was stronger in monkey G but only had a trend in monkey D (**Supplementary figure 13**). The CPD analyses at the level of individual neurons were also consistent (**Figure 6b, c**).

- b. We also assessed how well the DLPFC responses could be explained by the CV and the UCV. The average CPDs of the CV and the UCV were both significantly above chance level after the stimulus onset and similar during most of the trial period (**Figure 6d**). It was also confirmed by the regression model with the average firing rates (**Figure 6e, f**). The CPDs of the CV and the UCV were evenly distributed on both sides of the diagonal line, indicating that the UCV was encoded as well as the CV in DLPFC ($p=0.12$).

All these analyses were also performed in individual monkeys (**Supplementary figure 13**).

4. We did the same analyses for the DLPFC as we did for the OFC in **Figure 5** and showed the results in **Supplementary figures 14**. The DLPFC neuronal responses were similar when the monkeys were attending to either the cued or un-cued stimulus. This was true for both the positive-tuned neurons (attention location: $F_{1,674}=0.83$, $p=0.36$ in two-way ANOVA) and the negative-tuned neurons (attention location: $F_{1,724}=0.96$, $p=0.33$ in two-way ANOVA). In general, the results were very similar to what we observed for the OFC.

With these new analyses, we have revised the paper accordingly. The conclusion is that the value was encoded and modulated similarly in OFC and DLPFC, while the information on top-down spatial attention was only maintained in the DLPFC.

3) I am concerned with the terminology which is in many places confusing.

- The use of the word “salience” for reward value is unnecessary and confusing. Salience is used by different authors in different manners, causing a lot of confusion in the literature. For example, some use it for the stimulus-driven “attention-grabbing” ability of stimuli, others for task relevance, and here it is used for reward value. The authors should simply replace this term by “reward value” to avoid this confusion and thereby make the text more comprehensible. There is no need to use salience here.

We admit that the reviewer’s point is reasonable. However, we would like to keep the word “salience” to provide the contrast against the top-down attention cue in the task. In addition, the modulation on the OFC neurons responses was not graded as the word “value” would imply. It was almost an all-or-none effect. Using “salient” and “non-salient” provides this distinction. To reduce potential confusion, in the new revision we use the phrase “reward salience” and “top-down attention” to describe the two potential modulatory sources of value encoding.

- Same with “salient value” and “non-salient value”. Replace this by “higher value” and “lower value”, which is the more precise terminology.

Again, we prefer to keep the word “salience” to emphasize the all-or-none effect. By defining the SV and the NSV clearly early in Results, we hope that we have provided a clear description of the results.

- “Cued value” and “non-cued value” are also misnomers. The monkey is cued to attend an item that will change luminance. But this attention cue has no impact on the value, hence it does not make sense to talk about “cued value”. It is the shape of the two cues (and not their

luminance nor their locations) that jointly determine the expected value, because that is the average. Hence, the attention cue has no predictive value for the amount of reward that the monkey can expect to receive.

This is a fair point. The more accurate expressions would be “the value associated with the shape at the cued location” and “the value associated with the shape at the un-cued location”. However, we feel that these expressions would become wordy and repetitive when they appear in the writing frequently. We hope with proper and clear definitions, using CV and UCV consistently across the manuscript would be acceptable.

- The location cue is suggested to cue “top-down attention” and the shape cues “bottom-up” reward saliency. Why is one top-down and the other bottom-up? The spatial cue tells the monkey the most likely location of the luminance change that instructs the timing of an eye movement response. I.e. it is an instruction to attend a spatial location. The shape cues, in their combination, inform the monkey about the amount of the reward that the animal should expect after the trial. The information about both cues is presumably held in working memory for a while. The location cue needs to be in working memory because the monkey needs to remember where the luminance change will occur that is the go-cue for the eye movement response. The shape cue is in working memory because the monkey prepares for the amount of reward it will receive. Why would one cue be bottom-up and the other top-down as they are both in working memory? What is the logic here? I believe that both cues should be treated on equal footing and that the term “top-down” should be removed (everywhere).

The reviewer raised an important point. Attention is complicated. Attention from different sources can be transformed and combined. Yet, keeping the distinction between top-down and bottom-up is still useful in describing the source of attention and is commonly accepted in the field.

In our particular task, the location cue appeared at the opposite side of the luminance change. The monkeys had to direct their attention not to the location cue itself but the other side. This aspect of the task design ensures that the attention comes from the executive center of the brain and is top-down.

On the other hand, bottom-up cues can often lead to top-down attention, especially when they are useful for the behavior. In our task, however, we believe that the reward salience started from a bottom-up source due to the extensive training, but it was not relevant to the luminance change detection. Therefore, the monkeys would ignore them and not incorporate the signal into the “top-down” attention. If that is not the case, one would expect that the reward salience should affect the monkey’s behavior and lead to a significant performance enhancement at the location associated with the salient reward. This was not observed in both monkeys’ reaction times, and it was quite weak in their hit rates. The “top-down” location cue dominated the monkeys’ performance. Therefore, we don’t believe it is fair to put the attention based on the location-cue and the attention based on the reward salience on an equal footing.

Importantly, no matter whether one believes the reward salience is top-down or bottom-up, there is no denying that it has a smaller effect on the behavior but dominates the value encoding in the OFC. That is the central message of the paper.

Yet, we acknowledge the reviewer's point. We no longer use the word bottom-up when describing our findings and analyses in Results. We state our points in Discussion and explained why we consider reward salience is bottom-up, but otherwise describe it just as reward salience in Results.

- It might be useful to rewrite the section that includes a reference to Maunsell (2004 TiCS) who actually discussed that it is difficult, if not impossible, to dissociate reward from attention in NHP work. Indeed, in the present paradigm the spatial cue (frame) is important to receive reward (it is thereby also a sort of reward cue) because if the eye movement instructed by the luminance change at the cued location is too late or too early no reward will be obtained. The authors might want to revise their statement in line 463 that attention and reward can be dissociated in the brain, for two reasons. First, this statement is about spatial cues and shape cues, which in the present paradigm instruct or give information about different aspects of the task and are all necessary to get the reward at the end. The outcome would have been different with another design of the task. Second, if the dissociation is about the different roles of DPFC vs. OFC, I noted in my point 2 that this dissociation was not achieved convincingly.

The shape cues informed the monkeys of the potential rewards, which were not necessary for the monkeys to perform the task. The monkeys' decisions were not based on the values. One may even argue that they could be detrimental if the salient stimulus was not the one that would change illuminance. Nevertheless, the information was encoded by the OFC and dominated the OFC neurons value encoding.

The original purpose of the study focused on the attentional modulation of OFC's value encoding, and we only meant to use DLPFC recordings as evidence to demonstrate that the monkeys directed their attention according to the cue. Combined with the behavior data, the results excluded a potential explanation of why OFC neurons mainly encoded SV: the monkeys were wrongfully using the reward salience to do the task. Now seeing that the data we have allows us to do a more comprehensive comparison between the OFC and the DLPFC, we included such analyses in the manuscript.

John Maunsell actually read our drafts and provided a lot of feedback during the preparation of the manuscript. The statement about the dissociation of reward and attention (#463 in the original manuscript) was what he thought was the highlight of the study.

- To follow up on the last point, it seems that the authors' reasoning implies that there is only one type of attention and that if attention facilitates processing at a particular spatial location, then that is where attention is. However, there are many forms of attention, which can also be object based, feature based, etc. that all be expressed at the same time. This is another reason to keep the terminology closer to the experimental design and why I advise to use terms like "reward magnitude cue" and "spatial cue" rather than "top-down attention" and "bottom-up value". One example statement where this is an issue is in line 418 "The dominance of reward saliency that we observed cannot be attributed to top-down attention, decision making, or eye movements". That statement holds in the present paradigm for spatial attention to the most likely luminance change location, but it may be wrong for object-based attention and feature-based attention, which may be directed to shapes, in particular shapes that predict a high reward value. We know from previous work by the lab of e.g. Desimone that there are different circuits and neurons coding for shape and feature based attention. The

same problem occurs in line 461 suggestion that top-down attention cannot help very much. Help with what?

Thanks for the suggestion. We have made changes in the manuscript according to the suggestion. The sentence in original line #418 (#503 in the revision) now reads “The dominance of reward saliency that we observed cannot be attributed to *spatial* attention, decision making, or eye movements”. The sentence in original line #461 (#540 in the revision) now reads “...and top-down *spatial* attention, when directed toward the less salient stimulus, did not enhance its encoding very much in the OFC.”

4) The response of the OFC neurons was mainly driven by the stimulus with highest reward value and the value of the stimulus associated with less reward had little influence on activity (replicating an earlier study by the same team, Xie et al., 2018). One wonders whether this insensitivity to the lower reward value was also expressed in behavior. For example, did the reaction time or the probability that the monkeys aborted a trial (e.g. broke fixation) also not depend on the lower reward value?

This is a great question. We found that monkeys were sensitive to both the SV and the NSV behaviorally, but the effects were weak and inconsistent between monkeys. This creates contrast against OFC’s response patterns.

To examine whether the reaction time was influenced by the NSV, we regressed the averaged reaction time against the SV and the NSV. We found that both the SV and the NSV significantly influenced with the reaction time in monkey G (SV: $p=5.7e-102$; NSV: $p=1.2e-8$, t-test) but not in monkey D (SV: $p=4.8e-3$; NSV: $p=0.13$, t-test). This was consistent with what we’ve found before (**Figure 1 d, e**).

A similar analysis was also performed on the break fixation rate. Trials in which the fixation breaks happened after the stimulus onset and before the luminance change were analyzed. We found that both the SV and the NSV significantly contributed to the break fixation rate in monkey D (SV: $p=4.9e-138$; NSV: $p=7.6e-8$, t-test of the corresponding coefficients) but not in monkey G (SV: $p=8.2e-8$; NSV: $p=0.22$, t-test of the corresponding coefficients).

- Did the authors more generally look at the probability of fixation breaks and the speed of the saccade as markers of reward expectancy (see e.g. Takikawa et al. (2002) Modulation of saccadic eye movements by predicted reward outcome. Exp Brain Res 142:284–291)?

Thanks for the suggestion.

We looked at how fixation breaks correlated with the expected value, which is the average of the SV and the NSV. Again, we define the break fixation rate as the proportion of trials with fixation breaks between the stimulus onset and the luminance change. Linear regression analyses showed that the expected value significantly influenced the break fixation rate in both monkeys (monkey D: $F_{11,840}=87.42$, $p=5.0e-131$; monkey G: $F_{11,1126}=30.13$, $p=4.9e-56$, One-way ANOVA) but in different ways. Monkey D was more likely to break fixation in trials with larger expected values, whereas monkey G tended to break fixation when the expected value was small (**Figure R2**).

Figure R2. Break fixation rate under each expected reward. The error bars are SEM across sessions.

Next, we examined how the expected value modulated the saccade velocity. Following the procedure described by Takikawa⁵, a saccade was defined as when the eye movement velocity and acceleration exceeded the threshold values of 30°/s and 60°/s², respectively, the velocity must exceed 45°/s for at least 10 ms, and the total duration of the eye-movement is longer than 15 ms. The peak velocity of eye movement was calculated and regressed against the expected value. We did not find any correlation between the peak eye-movement velocity and the expected value in either monkey (monkey D: $F_{11,840}=0.07$, $p=1.0$; monkey G: $F_{11,908}=0.65$, $p=0.78$. One-way ANOVA).

5) The quality of the figures was not all that good, but I noticed that the cell of Fig. 3a start coding for reward expectancy before the stimuli appear on the screen. Is that real? I am curious because the same effect occurs in Fig. 3b. Or is that a smoothing effect? Please explain.

Figure 3a, b, c was an example neuron and only used same-stimulus trials as a baseline to measure tuning. They were only a small number of trials ($n=71$). The plot of the same neurons with different-stimulus trials, which were much more than same-stimulus trials, exhibits significant value encoding only after the stimulus onset (**Figure 3d**). In addition, the PSTHs in **Figure 3a, b, c** are smoothed with a Gaussian kernel, which combines the responses before and after. Therefore, signals may appear earlier than they actually do.

To further confirm that value was not significantly encoded before the stimulus onset, one-way ANOVA analyses were conducted at each 50-ms time bin (100-ms smooth window). We found that the first significant time point was around 150 ms after the stimulus onset (**Figure R3a**, $p<0.01$ with FDR corrections for multiple comparisons). Similar analyses were also conducted on the positive-tuned and negative-tuned OFC neurons (**Figure R3b, c**)

Figure R3. OFC responses to the same-value stimulus pairs. **a.** The responses of an example OFC neuron averaged at each 50-ms time bin (100-ms smooth window), aligned to the stimulus onset (left) and the luminance change (right). Trials were sorted for the same value from high (black curve) to low (lightest grey curve). Black segments above indicated when groups sorted for the same value were significantly different (one-way ANOVA, $p < 0.01$ with FDR corrections for multiple comparisons). Error bars indicate SEM across trials. **b.** The population response of the positively tuned OFC neurons ($n=74$). **c.** Same as **b**, but for the negatively tuned OFC neurons ($n=54$). Error bars indicate SEM across units in **b** and **c**.

6) It is odd that 0 drops in the paper actually correspond to 0.033ml (line 609). Hence 0 drops is also a bit misleading, why not simply putting reward magnitude in ml, to accurately describe what was done?

The reviewer's concern is reasonable. We originally did use 0 drops in the initial training but added a very small amount of juice to encourage the monkeys' performance during recordings. As the reward magnitudes in ml are not round numbers, we feel that it would be a bit wordy if we write 0.033 ml everywhere in the manuscript. To reflect the reviewer's concern, we now use ~0 to indicate the smallest reward magnitude in the revision.

7) It would be good to explain the regression analysis in lines 663 and following a bit better. If CV and UCV are predictors then adding CV-UCV as an extra predictor cannot work, because there is no variance that remains to be explained by the difference. Can the authors spell out how this was done?

The regression analyses described in section (#765) only include single variables. Correlations among these variables are ignored. Therefore, they only provide a general picture of how strongly each signal may be represented by the neurons. The number of neurons selective to each variable is likely an over-estimation. We provide a correlation table (**Supplementary table 1**) as a reference to help the readers to interpret the numbers.

The CPD analyses in the main figures include all relevant variables and address the issue of inter-dependency between the variables.

8) Supplementary Figs. 5 and 6 were missing from the Ms, and I have therefore not been able to judge the individual monkey results of these analyses.

We have updated the supplementary figures and double-checked all the figure references in the manuscript.

Smaller points:

Please make it very clear at the beginning of the results section that the location of the two shape cues does not convey any information about reward expectancy. It took me some time to realize that this is the case.

Thanks for the suggestion, now we add the following sentence in Results when we explain the task: "The initial frame cue, as well as the stimulus where the luminance occurred, did not convey information about which stimulus's reward would be delivered."

Line 77: "The monkeys had to direct their attention [...] to detect the luminance change". Really? How do the authors know that they could not detect the change without attention. This statement is erroneous because one monkey should completely ignore the luminance change at the non-cued location.

That's a fair point. We've rewritten the sentence as: "*To achieve a better performance, the monkeys had to direct their attention...*"

Line 86: What is "an early stage of reward processing"? Do the shape not have to be processed from early visual cortex up to the category level and then in OFC? If so, what is "early" about it?

The idea is that the raw value information represented in the early stage is processed somewhere in the brain, and only the behaviorally relevant value information would be

passed on to the later stage of processing. In our case, what is relevant should be the expected reward for a correct response, which is the mean value of the two stimuli. The OFC did not encode the mean value. It simply represented the value of the salient stimulus, which we suspect may come from the amygdala, regardless of the task demand.

We recognize the claim may be too speculative and have removed the sentence from the new revision.

In the discussion it would be useful to include a few lines explaining how the present results relate and go beyond Xie et al. 2018.

Previously in Xie et al. 2018, we found that reward salience modulated OFC's value encoding via an all-or-none mechanism, and we hypothesized that would be also true for top-down attention. This was, however, not tested in Xie et al., 2018. The new findings disprove this hypothesis.

In the last line of the first paragraph in Discussion, we wrote, "The results argue against the previously proposed theory that attention serves as a selection mechanism for the OFC's value encoding when multiple items with different value associations are presented".

Trials in which a high reward is at stake might cause a state of general attentiveness or arousal (19). Furthermore, reward value might influence "selective" attention; in the presence of multiple stimuli, attention might be attracted to those that are more rewarding. Studies on the neuronal correlates of expected value may therefore have measured attention shifts (29, 32).

In general, high-reward stimuli attract attention. However, in our task paradigm, the location indicated by the cue might direct attention to or away from the high-reward stimulus (SV). Both the behavior and the recording experiment results indicated that the location of attention was consistent with the cue, but not the SV location. With these results, we were able to evaluate how the value-encoding of the OFC was modulated by attention.

Typo on line 431: comes -> come

Fixed. Thanks.

Line 436: I disagree that the frame was "valueless" because it noted progress of the monkey through the trial and indicated that fixation has been attained for some time, that the trial had started and hence that a new reward was soon to be obtained. Same line: "bottom-up signal": not sure it was also registered in working memory and therefore would also have top-down properties.

We have revised the sentence as follows: "the onset of the frame, *although not providing value information on the trial's reward outcome*, was a strong bottom-up visual signal."

Reviewer #3 (Remarks to the Author):

It is well established that neural activity in the orbitofrontal cortex tracks the value of predictive stimuli. The current experiment tests whether this value coding is controlled by top-down attentional mechanisms or by the salience (learned value) of stimuli. Two monkeys were trained on a Posner-like task in which they had to respond to a luminance change in either of two stimuli presented on the left- and right-hand side of the screen. The side with the luminance change was predicted by an attentional cue (80 or 90% validity). The two stimuli were previously associated with different amounts of reward. Upon a correct response, either of the two reward magnitudes predicted by the stimuli were paid out (50% chance for either). The “salient stimulus” was defined as the stimulus with the higher value. Behavior (accuracy and response times (RT)) was affected by both the location of the attentional cue and the location of the salient stimulus. Activity of LPFC neurons tracked animals’ spatial attention, whereas activity of OFC neurons tracked the value of the salient stimulus, regardless of whether it was cued or non-cued, though there was a slight preference to encode the value of the cued stimulus.

The manuscript has several strengths. The question of what determines value encoding in the OFC is important and timely, and the neural recordings from LPFC to track animal’s attention is very nice. However, I have several major conceptual and methodological concerns that dampen my enthusiasm. Most importantly, I don’t think the conclusion that “the results strongly argue against the previously proposed theory that attention serves as a selection mechanism for the OFC’s value encoding” is justified.

Major concerns

1. My most important concern is that I don’t think that the behavioral task can provide meaningful insights into the mechanisms controlling value encoding in the OFC.

1.1. The reward delivered on a given trial was randomly selected from the two presented stimuli. But their value was otherwise irrelevant for guiding behavior. The only thing that mattered to obtain the reward was to detect and respond to the luminance change. Thus, the cue likely directed attention to the luminance of the stimuli not their shape. Based on this task design, it is unsurprising that attentional cues did not strongly affect value signals in OFC.

Indeed, the task is not a value-based decision-making task. This is actually one of the important design features for the following reasons.

1. Value-based decision-making tasks usually involve choosing the options associated with larger values. In addition, choice and attention are tightly coupled. The chosen option, along with its larger value, almost always has the animals’ attention in such a task. This is not ideal if we want to disentangle the effects of attention, value, eye movements, and decision making on neural activities.
2. Although value information is not necessary for the monkeys to perform the task, they were still working for the rewards, and the behavior showed that the rewards affected the monkeys’ performance (**Figure 1b, e, f**).
3. Many studies consistently show that the OFC encodes value in passive-viewing and other tasks in which animals did not have to actively use the value information, such as Xie et al⁴.

Therefore, we believe that the current design, although not testing the OFC's value encoding under the strongest attention condition, is a necessary compromise and is appropriate for our research question.

1.2. There is also no benefit of selectively paying more attention to the value of any of the two stimuli. Rather, the only potentially relevant value was the average value predicted by the two stimuli, as this signals the amount of reward the animal can expect. But whether OFC activity tracked the average value was not tested. And even if OFC codes the higher value of the stimuli, this could simply reflect an optimism bias.

The average value reflects both the SV and the NSV. If the neurons encoded the average value, one would expect the OFC to encode the SV and the NSV equally well, which was clearly not the case. Only the SV affected the OFC neurons' responses. This was also what was demonstrated in Xie et al⁴.

Nevertheless, we have added several new analyses to test whether the OFC activities encoded the average value. As the SV and the average value were highly correlated (**Supplementary table 1**), we calculated the coefficient of partial determination (CPD) with a regression model that contained both value variables as well as a variable for the frame cue location. If the OFC responses reflected the average value, we would expect that the CPDs of the average value were higher than or at least equaled the CPDs of the SV. However, this was not the case. As shown in **Figure R4** below, the CPDs of the SV were significantly higher than the CPDs of the average value, and the CPDs of the average value were not significant most of the time. As the information provided by this figure could be largely deduced from other existing analyses, we were inclined not to include it in the manuscript.

Figure R4. Encoding of the average value was weak in the OFC. Significance was assessed with two-tailed paired t-tests ($p < 0.005$, with FDR corrections for multiple comparisons) compared to a baseline computed with the average CPD between 0 and 200 ms before the cue onset and across different regressors. The blue, yellow, and green bars at the top indicate the significant CPDs of the cue location, the SV, and the average value, respectively, and the black

bar indicates the significant difference between the SV and the average value. The error bars indicate SEM across neurons.

1.3. Because the current task does not allow animals to engage in value-based decision-making of any sort, it is unclear why this would be a good way to study the role of OFC in value-based decision-making. There is no deliberation between different options like in the task used by Rich and Wallis, where animals could make choices between alternative options. There is simply no reason to do this in the current task where the expected reward on a given trial was independent of which stimulus was attended.

As we have stated above, having monkeys not to do value-based decisions is necessary for our study.

We'd like to point out that the proportion of value encoding neurons that we found in the OFC was comparable to that reported in studies of value-based decision-making tasks. As the reviewer correctly pointed out that the reward salience is actually not relevant in this task, it provides an even stronger contrast to its role in modulating value encoding in the prefrontal cortex.

Moreover, our results are actually compatible with the studies using decision-making tasks. For example, the SV (comparable to the chosen value of correct trials in Padoa-Schioppa's studies^{6,7}) was much better represented than the NSV (comparable to the un-chosen value) in the OFC, and the DLPFC neurons well reflected the SV location (comparable to the chosen direction of correct trials⁸). However, in those previous studies, the attention presumably is always consistent with the choice, thus also consistent with the SV, so one cannot tell whether their finding should be explained with the SV, attention, or choice. We provided a better way to tease apart the effects of these factors with a task not directly using value.

2. I am also not convinced that the main conclusions are well supported by the data.
2.1. Although more neurons tracked the value of the salient stimulus ($105/357 = 30\%$), a large number of OFC neurons ($34/357 = 10\%$) did track the value of the non-salient stimulus. This could suggest that different populations of OFC neurons track the value of the different stimuli. However, this is contrary to the authors' conclusion that OFC ignored the value of the non-salient stimulus.

As shown in **Figure 4b, c**, not only were the number of OFC neurons significantly encode NSV small, their CPDs were also small and clustered near 0. In addition, some of the NSV encodings could be attributed to the cue, which had a significant, although weak, effect on OFC's value encoding. In trials when the NSV is the attended cue, one also would expect some encoding of the NSV.

Nevertheless, in the new revision, we softened our language, so now the statement (line #540) reads: "The less salient stimuli are *largely* ignored by the OFC neurons."

2.2. Similarly, although OFC neurons tracked both the value of the cued and non-cued stimulus, at the individual neuron level, there was a preference for the cued over the non-cued values. Again, this is contrary to the main conclusion that top-down attention does not affect value coding in the OFC.

We actually stated that top-down attention weakly modulated OFC's value encoding. In particular, in Abstract, we wrote

*However, the OFC neurons' value encoding was dominated by the reward saliency associated with the stimuli and was only **weakly** modulated by the top-down attention.*

3. The experiment is fundamentally a two factorial design (salient vs. non-salient X cued vs. non-cued). Throughout the manuscript, the authors report the averages for each factor level across the other factor, rather than reporting the four individual cells separately. This would be necessary to see whether there are any interactions between the two attentional mechanisms.

3.1. Please add an analysis similar to Figure 3j,k,i, for all four conditions separately (salient-cued, non-salient-cued, salient-non-cued, non-salient-non-cued)

We have added the requested plot as **Supplementary figure 7**. The figure clearly shows that the OFC neurons' responses were dominated by the salient value.

3.2. Again, related to Figure 4, it would be important to show whether neurons track the salient and non-salient value separately for cued and non-cued locations. That is, please compute CPD for all four conditions (salient-cued, salient-non-cued, non-salient-cued, and non-salient-non-cued). This could show whether, for instance, the non-salient value is more strongly encoded when it is cued.

We have added the requested plot as **Supplementary figure 9**. The results provide further support to our conclusions, showing that the CPDs of the salient-cued value and the salient-un-cued value were significantly above chance level during the whole shape period. In contrast, we did not observe any significant encoding period for the non-salient-cued value and the non-salient-un-cued value.

3.3. Finally, it would be important to show and analyze behavior (Figure 1b-e) based on each of the four possible conditions (valid-salient, valid-non-salient, invalid-salient, invalid-non-salient) rather than the averages for valid/invalid and salient/non-salient.

We have provided a revised **Figure 1** in which all conditions were plotted separately. The performance difference between the valid and the invalid conditions is the largest, suggesting that the monkeys directed their attention to appropriate locations.

4. It is unclear what constitutes an incorrect response in this task. There was only one response target, and so animals had to simply indicate that any of the two stimuli had changed their luminance. This would mean that any response made on any trial was correct. So what was an incorrect response? If the monkey did not respond? Or if they responded too early or too late?

Monkey D was trained to detect a luminance change at either location, and it had to report it by making an upward eye movement within a time window between 100 and 400 ms after the luminance change and holding the fixation on the eye movement target for 300 ms (hold-target period). Failures in any of these requirements would be classified as errors. Errors include no responses within the response time window (miss trial), wrong saccade directions, and fixation breaks. The latter two were not included in any analyses. Monkey G was trained

to respond to the luminance change at the cued location only (target). In the trials with luminance change at the un-cued location (distractor), saccades toward the eye movement target within 100 to 400 ms after the luminance change are labeled as false alarms. The reason that we used a different task for monkey G is to reinforce its correct assignment of attention. Further details regarding the task difference can be found in the revised Methods and Results

References:

1. Posner, M. I. Orienting of attention. *Q. J. Exp. Psychol.* **32**, 3–25 (1980).
2. Tremblay, S., Pieper, F., Sachs, A. & Martinez-Trujillo, J. Attentional Filtering of Visual Information by Neuronal Ensembles in the Primate Lateral Prefrontal Cortex. *Neuron* **85**, 202–215 (2015).
3. McGinty, V. B. & Lupkin, S. M. Value signals in orbitofrontal cortex predict economic decisions on a trial-to-trial basis. *bioRxiv* 2021.03.11.434452 (2021).
4. Xie, Y., Nie, C. & Yang, T. Covert shift of attention modulates the value encoding in the orbitofrontal cortex. *Elife* **7**, e31507 (2018).
5. Takikawa, Y., Kawagoe, R., Itoh, H., Nakahara, H. & Hikosaka, O. Modulation of saccadic eye movements by predicted reward outcome. *Exp. Brain Res.* **142**, 284–291 (2002).
6. Padoa-Schioppa, C. & Assad, J. A. Neurons in the orbitofrontal cortex encode economic value. *Nature* **441**, 223–226 (2006).
7. Padoa-Schioppa, C. Neurobiology of economic choice: a good-based model. *Annu. Rev. Neurosci.* **34**, 333–359 (2011).
8. Lin, Z., Nie, C., Zhang, Y., Chen, Y. & Yang, T. Evidence accumulation for value computation in the prefrontal cortex during decision making. *Proc. Natl. Acad. Sci. U. S. A.* **117**, 30728–30737 (2020).

REVIEWER COMMENTS

Reviewer #1 (Remarks to the Author):

I have read the new additions and analysis the paper, which have made it stronger and more solid. I keep liking a lot the careful experimental design and the somehow surprising results.

Reviewer #2 (Remarks to the Author):

The authors substantially improved the Ms, but I remain concerned about the confusing terminology which should be fixed before I can support publication. The direct comparison between OFC and LPFC is also still missing.

1) It is distracting and unnecessary to replace the obvious terms "high-reward shape" and "low-reward shape" by "salient shape" and "non-salient shape", respectively. The use of the wording "stimulus salience" for "reward value" causes confusion in the Ms, and will make the story difficult to follow for many in the field.

2) I find the use of bottom-up attention and top-down attention still confusing. In their rebuttal the authors write: "On the other hand, bottom-up cues can often lead to top-down attention, especially when they are useful for the behavior. In our task, however, we believe that the reward salience started from a bottom-up source due to the extensive training, but it was not relevant to the luminance change detection. Therefore, the monkeys would ignore them and not incorporate the signal into the "top-down" attention."

- I would argue that the same is true for the spatial attention signal, which was driven by a bottom-up cue and then the location on the other side was attended.

The authors then continue "If that is not the case, one would expect that the reward salience should affect the monkey's behavior and lead to a significant performance enhancement at the location associated with the salient reward. This was not observed in both monkeys' reaction times, and it was quite weak in their hit rates. The "top-down" location cue dominated the monkeys' performance. Therefore, we don't believe it is fair to put the attention based on the location-cue and the attention based on the reward salience on an equal footing."

- However, they demonstrate that the reward magnitude influences reaction time and the probability of fixation breaks. Hence, it remains unclear why these effects are not also "top-down". Note that I am not disputing the relevance of the results, just a more careful wording would be appropriate.

They then end their reply by stating "Importantly, no matter whether one believes the reward salience is top-down or bottom-up, there is no denying that it has a smaller effect on the behavior but dominates the value encoding in the OFC. That is the central message of the paper."

- My point is that the description of the results is better if these "beliefs" should be taken out of the equation and replaced by more neutral terms.

- This problem is quite pervasive in the Ms, e.g. in the introduction, results and discussion

3) The authors now also carry out both analyses for the two areas and find that LPFC also codes for reward value. I am still missing the direct comparison between the results of the two areas. E.g. are regression coefficients for spatial attention larger in LPFC and those for reward in OFC?

- It might be better to keep the LPFC results for attention and reward magnitude in the same section. In the current description they are interrupted by the OFC results.

4) The authors state at many places in the Ms that the low-reward cue (they call it non-salient) has little effect on OFC activity. I was therefore a bit surprised by the clear influence in Supp Fig. 7c,d. Can you explain the discrepancy?

- The statement on line 555 "the lack of representation of NSV" is an overstatement.

Minor

Figure 1 has very small fonts, making it difficult to read

Reviewer #3 (Remarks to the Author):

Although I am still not fully convinced, I am open to the authors' arguments for why their task is an acceptable compromise for answering basic questions about how attention affects value coding independent of choice.

Besides this, the authors adequately addressed my concerns and the new plots and analyses provide additional support for the authors' conclusions. I have no further comments.

REVIEWER COMMENTS

Reviewer #1 (Remarks to the Author):

I have read the new additions and analysis the paper, which have made it stronger and more solid. I keep liking a lot the careful experimental design and the somehow surprising results.

Thank you very much for reviewing our submission and helping us to improve the manuscript.

Reviewer #2 (Remarks to the Author):

The authors substantially improved the Ms, but I remain concerned about the confusing terminology which should be fixed before I can support publication. The direct comparison between OFC and LPFC is also still missing.

We would like to thank the reviewer for the positive comments and the constructive feedback. In the new revision, we have added the direct comparison between OFC and LPFC. We have also revised the manuscript to accommodate the reviewer's concern. In particular, we now use "spatial attention" to describe the results, and only use top-down and bottom-up when we speculate the source of the attention.

1) It is distracting and unnecessary to replace the obvious terms "high-reward shape" and "low-reward shape" by "salient shape" and "non-salient shape", respectively. The use of the wording "stimulus salience" for "reward value" causes confusion in the Ms, and will make the story difficult to follow for many in the field.

Thanks for the suggestion. In the new revision, we consistently use the word "reward salience" throughout the manuscript when describing the results to emphasize the type of salience signals we are studying in this paper. However, we continue believing "salient/non-salient" is a better choice of words than "high/low-reward" here for the following reasons.

- a. Just as physical (visual) salience, reward salience is a potential source of spatial attention. In Discussion, we point out that both reward and physical salience signals are reflected in the OFC. Replacing "reward salience" with "reward value" would lose the parallelism between the two.
- b. We demonstrate that the value-tuning of OFC neurons depended on which of the two stimuli was salient. As long as a stimulus is a salient one, it dominates the OFC neurons' value tuning, no matter what its actual reward value is. Using the distinction between salient and non-salient creates a better dichotomy than high/low reward value. Replacing the word "salience" with reward value would mislead readers into thinking the opposite.

- c. The term “reward salience” (or “incentive/motivational salience”) is commonly used in the field (eg. Ghazizadeh et al., 2016; Kahnt and Tobler, 2017; Kim et al., 2020). We follow the conventional definition and usage of this term and do not believe the way we use the term would cause confusion, especially with a clear definition of provided in the paper.

For these reasons, we prefer to keep the current terminology.

References:

- Ghazizadeh A, Griggs W, Hikosaka O (2016) Ecological origins of object salience: Reward, uncertainty, aversiveness, and novelty. *Front Neurosci* 10.
- Kahnt T, Tobler PN (2017) Reward, Value, and Salience. *Decis Neurosci An Integr Perspect*:109–120.
- Kim HF, Griggs WS, Hikosaka O (2020) Long-Term Value Memory in the Primate Posterior Thalamus for Fast Automatic Action. *Curr Biol* 30:2901-2911.e3.

2) I find the use of bottom-up attention and top-down attention still confusing. In their rebuttal the authors write: “On the other hand, bottom-up cues can often lead to top-down attention, especially when they are useful for the behavior. In our task, however, we believe that the reward salience started from a bottom-up source due to the extensive training, but it was not relevant to the luminance change detection. Therefore, the monkeys would ignore them and not incorporate the signal into the “top-down” attention.”

- I would argue that the same is true for the spatial attention signal, which was driven by a bottom-up cue and then the location on the other side was attended.

The authors then continue “If that is not the case, one would expect that the reward salience should affect the monkey’s behavior and lead to a significant performance enhancement at the location associated with the salient reward. This was not observed in both monkeys’ reaction times, and it was quite weak in their hit rates. The “top-down” location cue dominated the monkeys’ performance. Therefore, we don’t believe it is fair to put the attention based on the location-cue and the attention based on the reward salience on an equal footing.”

- However, they demonstrate that the reward magnitude influences reaction time and the probability of fixation breaks. Hence, it remains unclear why these effects are not also “top-down”. Note that I am not disputing the relevance of the results, just a more careful wording would be appropriate.

They then end their reply by stating “Importantly, no matter whether one believes the reward salience is top-down or bottom-up, there is no denying that it has a smaller effect on the behavior but dominates the value encoding in the OFC. That is the

central message of the paper.”

- My point is that the description of the results is better if these “beliefs” should be taking out of the equation and replaced by more neutral terms.

- This problem is quite pervasive in the Ms, e.g. in the introduction, results and discussion

To reduce the confusions, we now use “spatial attention” throughout the paper when we describe our findings, and only use top-down and bottom-up when we speculate the source of the attention.

3) The authors now also carry out both analyses for the two areas and find that LPFC also codes for reward value. I am still missing the direct comparison between the results of the two areas. E.g. are regression coefficients for spatial attention larger in LPFC and those for reward in OFC?

Thanks for your suggestion.

To directly compare spatial attention and value signals between the DLPFC and the OFC, we have add two new figures (**Figure 7** and **Supplementary figure 14**). We calculated the CPD of the attention cue location and the CPD of the SV with the average neuronal responses between attention cue-off and the luminance change for each region. The results are plotted in **Figure 7** (**Supplementary figure 14** for individual monkeys). Consistent with other results, the neuronal responses in the OFC were dominated by the SV but not spatial attention, while LPFC neurons encoded both SV and spatial attention.

We also carried out additional analyses to further confirm the results. In the figure attached below (**Figure R1**), we calculated the selectivity index ((Wu et al., 2020)) of the attention cue location and the SV separately for individual neurons. The selectivity index was defined as $SI = 2(AuROC - 0.5)$, where the AuROC is the area under the ROC curve. For spatial attention, the ROC curve measures the discriminability between left and right attention cue conditions based on trial-by-trial spike counts. For reward salience, the ROC curve measures the discriminability between SV and NSV. The significance of the SI for each neuron was determined by a two-tailed Wilcoxon rank-sum test (not corrected for multiple comparisons). We found that 25.8% of OFC neurons were selective to the SV, while only 6.59% of them responded to the attention cue location. In the DLPFC, there were comparable numbers of neurons tuned to the SV (22.2%) and to the attention cue location (18.7%). As this analysis is mostly consistent with the other figures and provides limited extra information, the figure is not included in the manuscript and only provided here.

Reference:

Figure R1. Selectivity indices of OFC and DLPFC neurons for the attention cue location and the SV. **a.** Distribution of selectivity indices. The orange bars indicate the OFC neurons that are significantly selective to the attention cue location or the SV. The magenta bars indicate the significantly selective DLPFC neurons. **b.** The proportion of OFC (orange) and DLPFC (magenta) neurons with significant selectivity to the attention cue location and the SV. **a, b:** monkey combined; **c, d:** monkey D; **e, f:** monkey G.

- It might be better to keep the LPFC results for attention and reward magnitude in the same section. In the current description they are interrupted by the OFC results.

Thanks for your suggestion. Considering our main goal is to examine how top-down attention modulates value encoding in OFC, we prefer to keep the current manuscript structure to highlight the OFC results. We took the advice from the reviewers and included the analyses on DLPFC's value encoding, which provides a nice comparison to the OFC results.

4) The authors state at many places in the Ms that the low-reward cue (they call it non-salient) has little effect on OFC activity. I was therefore a bit surprised by the

clear influence in Supp Fig. 7c,d. Can you explain the discrepancy?

The reviewer made an accurate observation. **Supplementary figure 7** looks like OFC neuronal responses were correlated with the NSV (the green and the purple line). However, this is because the average SV increases as the NSV increases. For example, the group “NSV=2” is composed of reward pairs of “2-4” and “2-8”, while the group “NSV=4” is composed of the reward pair of “4-8”. Therefore, the increase in SV may fully account for the trend observed in **Supplementary figure 7c, d**.

In **Supplementary figure 4**, we conditioned the SV and compared the neuronal responses under the non-salient values. It is clear that the tuning curves are mostly flat, indicating that NSV did not have a big effect on the OFC’s activity.

To help the readers to understand these fine points, we have labeled the reward pairs that contribute to each data point in **Supplementary figure 7**.

- The statement on line 555 “the lack of representation of NSV” is an overstatement.

We’ve revised the sentence to: “*the weak representation of NSV...*” in the new revision (line #571).

Minor

Figure 1 has very small fonts, making it difficult to read

Fixed.

Reviewer #3 (Remarks to the Author):

Although I am still not fully convinced, I am open to the authors’ arguments for why their task is an acceptable compromise for answering basic questions about how attention affects value coding independent of choice.

Besides this, the authors adequately addressed my concerns and the new plots and analyses provide additional support for the authors' conclusions. I have no further comments.

Thank you very much for being open and providing constructive comments for us to improve the paper.

REVIEWERS' COMMENTS

Reviewer #2 (Remarks to the Author):

The revision is excellent and I appreciate the inclusion of the direct comparison between OFC and LPFC neurons (the new Fig. 7) as well as the careful presentation of the results of the individual monkeys. Against my advice, the authors decided to stick with the confusing nomenclature of 'salient' for 'high-reward' and 'non-salient' for 'low-reward'. This is not ideal, but I prefer to not further delay publication of this very interesting set of findings.

Minor points:

- the authors state in line 582: "Our results suggest that the representations of attention and reward are dissociable in the brain." The more precise statement would be "the representation of spatial attention and reward expectancy based on shape information are dissociable in the brain".
- Is it conceivable that the differences between the DPFC neurons between the two monkeys as visible in e.g. Fig. S13 are caused by slightly different parts of DPFC that were targeted in the two animals, as can be seen in Fig. S1?

REVIEWERS' COMMENTS

Reviewer #2 (Remarks to the Author):

The revision is excellent and I appreciate the inclusion of the direct comparison between OFC and LPFC neurons (the new Fig. 7) as well as the careful presentation of the results of the individual monkeys. Against my advice, the authors decided to stick with the confusing nomenclature of 'salient' for 'high-reward' and 'non-salient' for 'low-reward'. This is not ideal, but I prefer to not further delay publication of this very interesting set of findings.

Minor points:

- the authors state in line 582: "Our results suggest that the representations of attention and reward are dissociable in the brain." The more precise statement would be "the representation of spatial attention and reward expectancy based on shape information are dissociable in the brain".

Thank you for your suggestion. We've revised the sentence to reflect the suggestion (line #426).

- Is it conceivable that the differences between the DPFC neurons between the two monkeys as visible in e.g. Fig. S13 are caused by slightly different parts of DPFC that were targeted in the two animals, as can be seen in Fig. S1?

To examine whether the differences in DLPFC responses between the two monkeys are caused by the discrepancy in their recording sites, we performed an analysis in which only neurons from the same subarea (area 46), and the results were similar to that in Supplementary Fig. 13. The current dataset cannot explain the difference between the two monkeys.